# Whole-exome sequencing analysis identifies risk genes for schizophrenia

Sophie L. Chick[1,2,3], Peter Holmans [1,2,3,4], Darren Cameron [1,2,3], Detelina Grozeva [2,3,5], Rebecca Sims [1,2,3,6], Julie Williams [2,3,4,6], Nicholas J. Bray [1,2,3], Michael J. Owen [1,2,3], Michael C. O'Donovan [1,2,3], James T. R. Walters [1,2,3] & Elliott Rees [1,2,3] ✉

Rare coding variants across many genes contribute to schizophrenia liability, but they have only been implicated in 12 genes at exome-wide levels of significance. To increase power for gene discovery, we analyse exome-sequencing data for rare coding variants in a new sample of 4650 schizophrenia cases and 5719 controls, and combine these with published sequencing data for a total of 28,898 cases, 103,041 controls and 3444 proband-parent trios. We identify associations for *STAG1* and *ZNF136* at exome-wide significance, genes that were previously implicated in schizophrenia by the SCHEMA study at a false discovery rate of 5%. We also find associations at a false discovery rate of 5% for six genes that did not pass this statistical threshold in the SCHEMA study (*SLC6A1*, *PCLO*, *ZMYND11*, *BSCL2*, *KLC1* and *CGREF1*). Among these genes, *SLC6A1* and *KLC1* are associated with damaging missense variants alone. *STAG1*, *SLC6A1*, *ZMYND11 and CGREF1* are also enriched for rare coding variants in other developmental and psychiatric disorders. Moreover, *STAG1* and *KLC1* have fine-mapped common variant signals in schizophrenia. These findings provide insights into the neurobiology of schizophrenia, including further evidence suggesting an aetiological role for disrupted chromatin organisation.

Schizophrenia is a severe and heterogeneous psychiatric syndrome, characterised by behavioural and cognitive symptoms which may be lifelong and are often inadequately resolved by antipsychotic medications[1,2]. Molecular genetic studies have established that the genetic architecture of schizophrenia is highly polygenic, with thousands of alleles of different frequencies from across the genome contributing to liability[3]. Common variants (minor allele frequency (MAF) > 0.01) measured on single-nucleotide polymorphism (SNP) genotyping arrays currently explain 24% of the variance in schizophrenia liability in European populations, and the largest genome-wide association study (GWAS) to date has identified 287 common variant loci at genome-wide significance (mean odds ratio (OR) 1.06; range 1.04–1.23)[4]. Statistical and functional fine-mapping approaches used in that report prioritised 106 credible causal protein-coding genes within the associated loci[4], although few of these genes can be considered definitively implicated.

Current estimates suggest that around 5% of variance in schizophrenia liability is explained by rare (MAF < 0.01) copy number variants (CNVs) and ultra-rare (MAF < 2 × 10⁻⁵) coding variants[3,5]. To date, 13 rare CNVs have been identified as schizophrenia risk factors, with

¹Centre for Neuropsychiatric Genetics and Genomics, Division of Psychological Medicine and Clinical Neurosciences, Cardiff University, Cardiff, UK. ²Division of Psychological Medicine and Clinical Neurosciences, Cardiff University, Cardiff, UK. ³Neuroscience and Mental Health Innovation Institute, Cardiff University, Cardiff, UK. ⁴MRC UK Dementia Research Institute, School of Medicine, Cardiff University, Cardiff, UK. ⁵Centre for Trials Research, College of Biomedical and Life Sciences, Cardiff University, Cardiff, UK. ⁶Moondance Dementia Research Laboratory, Cardiff University, Cardiff, UK. ✉e-mail: ReesEG@cardiff.ac.uk

ORs ranging between 1.8 and 81.2[6,7]. All but one of these overlap multiple genes, making it difficult to implicate those with causal effects. The exception involves non-recurrent, intragenic deletions of *NRXN1*, which encodes a pre-synaptic cell adhesion protein.

Rare coding variants (RCVs) contributing to schizophrenia are concentrated among a set of around 3000 genes known to be under selective constraint in humans against stop-gain, essential splice site, and frameshift mutations, collectively termed protein-truncating variants (PTVs)[8–11]. The Schizophrenia Exome-Sequencing Meta-Analysis (SCHEMA) consortium analysed data from 24,248 cases, 97,322 controls and 3402 trios, and identified a set of 10 exome-wide significant genes, with ORs ranging between 3 and 50, and a further 22 genes at a false discovery rate (FDR) of <5%[12]. A subsequent study by the Psychiatric Genomics Consortium (PGC) sequenced the coding regions of 161 genes in 11,580 cases and 10,555 controls, and meta-analysed the gene-level P-values for PTVs with those published in the SCHEMA study[12] to implicate two additional genes, *AKAP11* and *SRRM2*, at exome-wide significance[13]. Among the 12 genes currently implicated in schizophrenia at exome-wide significance, 5 are also enriched for RCVs in other developmental disorders (DDs), including intellectual disability[12,13]. Moreover, two genes, *GRIN2A* and *SP4*, were prioritised as credible causal candidates in the most recent large schizophrenia GWAS[4].

The functional roles of genes enriched for RCVs in schizophrenia have informed understanding of some of the disease mechanisms that may contribute to this disorder. While the genes currently implicated at exome-wide significance have diverse biological functions, several have roles related to glutamatergic synaptic signalling and transcriptional regulation[12,13]. Identifying additional genes enriched for RCVs in schizophrenia will provide further opportunities to understand the complex neurobiology underlying this disorder.

Here, we report whole-exome sequencing data in a new sample of 4650 schizophrenia cases and 5719 controls, and meta-analyse these with published exome-wide sequencing data for a total of 28,898 cases, 103,041 controls and 3444 proband-parent trios. In the largest exome-sequencing meta-analysis of schizophrenia to date, we report

exome-wide significant associations for *STAG1* and *ZNF136*, genes that were previously implicated in schizophrenia at FDR < 5% by the SCHEMA study[12]. We also find associations at FDR < 5% for six additional genes that did not reach this level of statistical significance in the SCHEMA study[12].

## Results

### New exome sequencing sample

We generated and analysed exome-sequencing data in 4650 schizophrenia cases and 5719 controls passing quality control, none of which have contributed to previous exome-sequencing studies of schizophrenia (Supplementary Methods). In this new sample, singleton (minor allele count (MAC) = 1 and absent in gnomAD controls[10]) PTVs, as well as singleton missense variants with a 'missense badness, Polyphen-2 and constraint' (MPC) score > 3, in constrained genes were enriched in cases compared with controls (Fig. 1, Supplementary Table 5). The effect sizes for these variants in the new sample were consistent with those in the published SCHEMA study (Fig. 1)[12]. The new cases were not significantly enriched compared with controls for singleton missense variants with MPC scores between 2 and 3 in constrained genes, but the effect size for these variants was again consistent with the SCHEMA study (Fig. 1).

The rate of singleton synonymous variants in constrained genes was significantly higher in the new controls than in cases (Fig. 1, Supplementary Table 5). In non-constrained genes, the rates of singleton PTVs and damaging missense variants (MPC > 3 and MPC 2-3) did not differ between cases and controls, but controls were still enriched for singleton synonymous variants (odds ratio (95% confidence interval) = 0.97 (0.95–0.99); P = 0.011). We tested whether the excess of singleton synonymous variants in controls is explained by greater sequencing depth in the controls (mean genotype depth = 34.2x) compared with cases (mean genotype depth = 26.0x). Sequencing depth in the new sample was positively correlated with the rate of singleton coding variants (Supplementary Note 1). We also performed a gene-set sensitivity analysis using control data that is matched with the cases for sequencing coverage. Here, the rate of synonymous singletons in

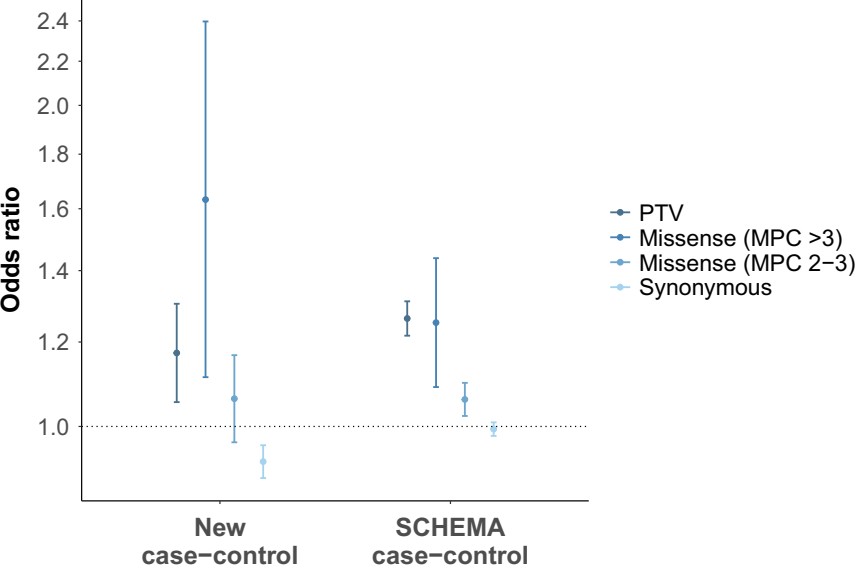

**Fig. 1 | Case-control gene-set analysis of singleton coding variants in constrained genes.** Constrained genes are defined as those with pLi scores ≥ 0.9 in gnomAD (*n* genes = 3051)[10]. The measure of centre is the odds ratio value and error bars denote 95% confidence intervals of odds ratio estimates. Odds ratios are plotted on a log scale. Odds ratios in the new case-control sample were derived from Firth's logistic regression models (Supplementary Methods). Published odds ratios from the SCHEMA sample were taken from Singh et al. (2022). PTV protein-truncating variants, MPC 'missense badness, Polyphen-2 and constraint' score, pLi probability of being loss of function intolerant. Source data for Fig. 1 are provided as a Source Data file.

**Table 1 | Novel schizophrenia exome-wide significant and false discovery rate <5% genes**

| Gene symbol | Variant class | SCHEMA case-control re-analysis | | Combined case-control analysis | | Trios | Case-control-de novo variant meta-analysis | |
|---|---|---|---|---|---|---|---|---|
| | | OR (95% CI) | P-val | OR | P-val | De novo variant count | P-val | Q-val |
| STAG1 | PTV + MPC > 2 | 4.9 (2.6–9.3) | **$4.6 \times 10^{-7a}$** | 5.6 (2.9–10.6) | **$3.7 \times 10^{-8}$** | 0 | **$3.7 \times 10^{-8}$** | $1.4 \times 10^{-4}$ |
| ZNF136 | PTV | 4.0 (2.2–7.5) | $3.8 \times 10^{-6}$ | 4.4 (2.4–8.1) | **$6.2 \times 10^{-7}$** | 0 | **$6.2 \times 10^{-7}$** | 0.0016 |
| SLC6A1 | MPC > 2 | 2.5 (1.3–5.0) | $1.0 \times 10^{-2}$ | 2.9 (1.5–5.6) | $2.2 \times 10^{-3}$ | 3 | $1.9 \times 10^{-6}$ | 0.0035 |
| PCLO | PTV | 4.0 (2.0–7.9) | $5.8 \times 10^{-5}$ | 4.5 (2.3–8.6) | $2.7 \times 10^{-6}$ | 0 | $2.7 \times 10^{-6}$ | 0.0043 |
| ZMYND11 | PTV | 7.1 (1.9–27.1) | $1.4 \times 10^{-4}$ | 7.8 (2.0–29.5) | $7.9 \times 10^{-5}$ | 1 | $1.6 \times 10^{-5}$ | 0.017 |
| BSCL2 | PTV | 3.9 (1.8–8.3) | $1.3 \times 10^{-4}$ | 4.5 (2.1–9.4) | $1.9 \times 10^{-5}$ | 0 | $1.9 \times 10^{-5}$ | 0.019 |
| KLC1 | MPC > 2 | 6.3 (2.5–15.7) | $1.1 \times 10^{-5}$ | 5.0 (2.1–11.9) | $5.0 \times 10^{-5}$ | 0 | $5.0 \times 10^{-5}$ | 0.042 |
| CGREF1 | PTV | 3.0 (1.7–5.3) | $1.6 \times 10^{-4}$ | 3.2 (1.8–5.6) | $5.0 \times 10^{-5}$ | 0 | $5.0 \times 10^{-5}$ | 0.042 |

CI confidence interval, PTV protein-truncating variant, MPC 'missense badness, Polyphen-2 and constraint' scores for missense variants.

P-values are unadjusted for multiple comparisons, and bold text indicates P-values exceeding Bonferroni significance ($P < 1.63 \times 10^{-6}$). Case-control P values and ORs were calculated using two-sided Cochran–Mantel–Haenszel tests. Case-control-de novo variant meta-analysis P values were calculated using Fisher's combined probability tests (see Methods for further details). Q-values show adjusted meta-analysis P-values using the false discovery rate approach.

[a]We note that STAG1 is exome-wide significant in our re-analysis of PTVs and missense (MPC > 2) variants in the SCHEMA case-control data alone ($P = 4.6 \times 10^{-7}$; Table 1), but it did not achieve exome-wide significance when we meta-analysed P-values for PTVs + missense (MPC > 3) and missense variants (MPC 2-3) using the weighting scheme adopted by the SCHEMA study ($P = 1.8 \times 10^{-5}$). The odds ratios (ORs) and P-values correspond to the variant class shown.

constrained genes remained higher in controls than in cases, but the rate of synonymous singletons in non-constrained genes did not differ between cases and controls (odds ratio (95% confidence interval) = 0.99 (0.97–1.01); $P = 0.36$). Moreover, the excess in cases of singleton PTVs and missense variants with MPC scores > 3 in constrained genes was greater in the sensitivity analysis when compared with the original analysis, suggesting that the higher control sequence coverage in the original dataset leads to conservative estimates of schizophrenia rare variant enrichment, rather than false positive discovery.

## Discovery of schizophrenia genes

We performed a per-gene RCV meta-analysis using data from the new case-control sample and published variant-level data from SCHEMA cases and controls (full details in Methods and Supplementary Methods), giving a total of 28,898 cases and 103,041 controls. Genes were evaluated for RCV enrichment using two-sided Cochran-Mantel-Haenszel (CMH) tests, with separate contingency tables for samples grouped by genetically inferred ancestry and exome capture platform. This analysis focused on variants with a MAC ≤ 5 in all cases and controls (SCHEMA + new sample) and ≤ 5 in 60,146 gnomAD controls[10], which is similar to the MAC threshold used in the SCHEMA study[12]. Single gene enrichment tests were performed for four variant classes: PTVs; PTVs + missense MPC > 3; PTVs + missense MPC > 2; missense MPC > 2. Similar to the approach used in the SCHEMA study[12], genes achieving a case-control CMH $P < 0.01$ were meta-analysed with gene-based de novo coding variant P-values derived from 3444 published schizophrenia trios[11], using Fisher's combined probability tests. In total, 30,674 single gene tests were performed, corresponding to an exome-wide significance threshold of $P = 1.63 \times 10^{-6}$ after Bonferroni correction. See Methods for a full description of our gene discovery approach.

We identified two novel risk genes at exome-wide significance (Table 1): STAG1 was associated with rare PTVs and missense (MPC > 2) variants ($P = 3.7 \times 10^{-8}$), and ZNF136 was associated with rare PTVs ($P = 6.2 \times 10^{-7}$). Both STAG1 and ZNF136 were previously implicated in schizophrenia by the SCHEMA study at FDR < 5%[12], with further support for these associations now provided in the new case-control sample. We also report six additional novel risk genes at FDR < 5% (Table 1). Among these, SLC6A1 and KLC1 are associated with schizophrenia based on evidence from missense variants (MPC > 2) alone. In an earlier study[11], we reported that cases were enriched for de novo missense variants (MPC > 2) in SLC6A1 (3 observed, 0.069 expected,

$P = 5.2 \times 10^{-5}$). The present study, with inclusion of case-control data, provides further support for association between missense variants in SLC6A1 and schizophrenia (Table 1). A literature review of the functions of the novel exome-wide significant and FDR < 5% genes reported in this study is presented in the Discussion and in Supplementary Note 2. Analysis of the spatio-temporal expression of the novel exome-wide significant and FDR < 5% genes is presented in Supplementary Note 3.

All variants in the new case-control sample that contribute to the associations reported in Table 1 are presented in Supplementary Data 1. The combined case-control and de novo variant meta-analysis results for all genes is available in Supplementary Data 2.

## Convergence of RCV-enriched genes in schizophrenia common variant and CNV loci

STAG1 and KLC1 overlap genomic loci with genome-wide significant common variant associations in the largest schizophrenia GWAS to date[4]. STAG1 narrowly failed the conservative criteria adopted in that study for prioritising a gene as likely to be causal. STAG1 was also reported as a likely RCV associated gene, since it was in the FDR < 5% set of the SCHEMA study[12]. Given the suggestive evidence implicating this gene by two independent study designs, STAG1 was considered by both studies to be a highly likely schizophrenia causal gene[4,12]. Our study strengthens the evidence implicating STAG1, with enrichment for rare PTVs and missense (MPC > 2) variants now attaining exome-wide significance. Like STAG1, KLC1 resides within a schizophrenia GWAS locus containing multiple genes (Fig. 2), but in that instance, the 95% credible set of causal variants was dispersed across 23 genes[4]; by implicating missense variants in KLC1 in schizophrenia at FDR < 5%, our study supports the prioritisation of this gene within its GWAS locus.

For genes overlapping the critical regions of previously implicated schizophrenia CNV loci (Supplementary Table 6), five were enriched for RCVs with nominal levels of significance ($P_{\text{uncorrected}}$ <0.05) (Supplementary Table 7). Here, the most significant RCV association was observed for PTVs in NRXN1, which is associated with non-recurrent single-gene deletions in schizophrenia. This association survived Bonferroni correction for the number of gene tests at this CNV locus ($n = 3$; $P_{\text{corrected}} = 0.00087$). Among multigenic schizophrenia CNV loci, the most significant RCV association was observed for PTVs in C22orf39, which overlaps the 22q11.2 deletion locus. Since none of the genes overlapping multi-genic schizophrenia CNVs were enriched for RCVs after Bonferroni correction for the number of gene

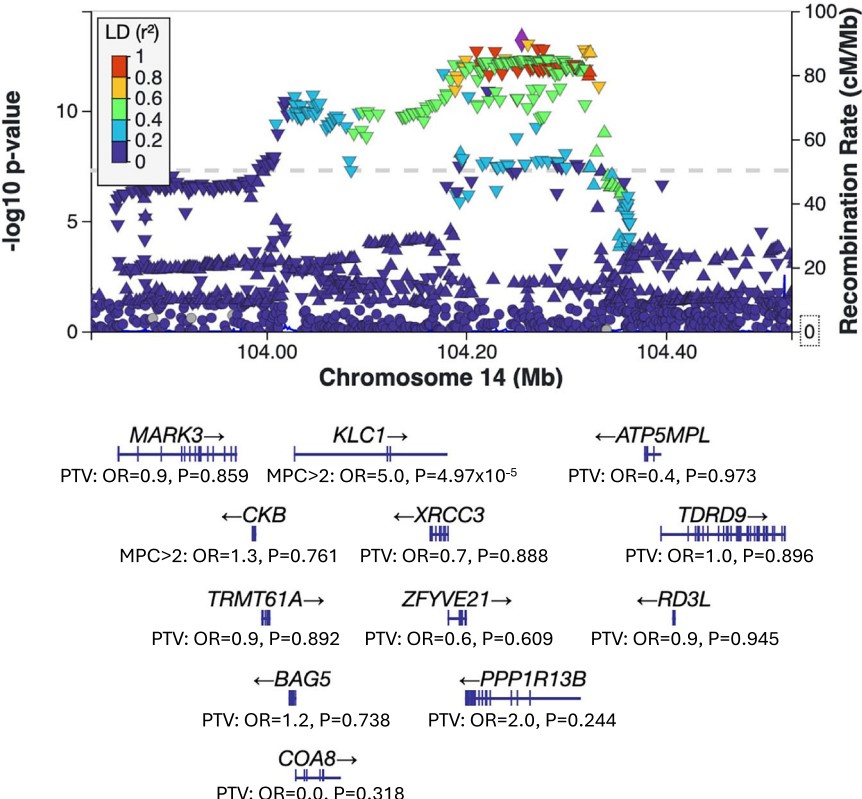

**Fig. 2 | Schizophrenia GWAS locus plot for *KLC1*.** Summary statistics for the PGC3 schizophrenia GWAS[4] were downloaded from the PGC website (https://pgc.unc.edu/for-researchers/download-results/) and visualised using LocusZoom[57]. The *y* axis shows unadjusted two-sided *P* values of each SNP from the PGC3 schizophrenia GWAS summary statistics[4], and the colour of each triangle shows the linkage disequilibrium with the index SNP. Shown below are the unadjusted two-sided *P*-values and odds ratios (OR) for each gene within 200KB of *KLC1*. *P*-values are from the case-control-de novo meta-analysis and ORs are from the Cochran–Mantel–Haenszel case-control analysis. *P* values and ORs are shown for the class of variant most strongly associated with schizophrenia.

tests in the given CNV locus (Supplementary Table 7), further evidence is needed to identify those with causal effects.

## Genic pleiotropic effects of schizophrenia genes

Genes enriched for RCVs are known to overlap between schizophrenia and other psychiatric and developmental disorders. We therefore examined whether the novel exome-wide significant and FDR < 5% genes identified in the current study are associated with RCVs in published studies of DD[14], autism spectrum disorder (ASD)[15], epilepsy[16] and bipolar disorder (BD)[17], as this would provide evidence of genic pleiotropy and further support their role in schizophrenia. There is evidence of genic pleiotropic effects for 4 of the 8 schizophrenia genes examined (Table 2). In *SLC6A1*, missense variants have broad effects across schizophrenia, ASD, DD and epilepsy, and PTVs are additionally associated with DD (Table 2). In *STAG1*, both PTVs and missense variants are associated with schizophrenia and DD (Table 2). In *ZMYND11*, PTVs are associated with schizophrenia, DD and ASD, whereas missense variants are associated with DD (Table 2). Finally, both schizophrenia and autism show evidence for association with PTVs in *CGREF1* (Table 2).

## Analysis of previously implicated genes in the new case-control sample

We analysed genes previously associated with RCVs in schizophrenia in the new case-control sample alone. When tested as a set, the 12 previously implicated exome-wide significant genes were enriched for rare PTVs (MAC ≤ 5 in the new sample and ≤5 in 60,146 gnomAD controls[10]) in the new cases compared with controls (OR = 4.97, $P = 2.7 \times 10^{-4}$; Supplementary Table 8). The 20 additional previously implicated genes at FDR < 5% were also enriched for rare PTVs in the new cases (OR = 2.48, $P = 1.6 \times 10^{-2}$; Supplementary Table 8). The effect sizes for PTVs in these previously implicated gene sets were significantly greater than the general effects observed for PTVs in constrained genes (Supplementary Table 9). The burden of rare missense variants and synonymous variants in previously implicated gene sets did not differ between the new cases and controls (Supplementary Table 8).

We then performed single-gene enrichment tests in the new sample alone for the 12 genes previously implicated in schizophrenia at exome-wide significance. Eight of the 12 genes had a higher burden of RCVs in the new cases compared to controls (OR > 1), three with a one-sided $P \leq 0.05$ (Supplementary Table 10).

## Discussion

To discover schizophrenia risk genes, we generated exome-sequencing data on a new sample of 4650 cases and 5719 controls, and meta-analysed these data with RCVs from published sequencing studies of schizophrenia for a total of 28,898 cases, 103,041 controls and 3444 trios. This represents, to our knowledge, the largest whole-exome sequencing study of schizophrenia to date.

We report association at exome-wide significance between rare PTVs and damaging missense variants in *STAG1* and schizophrenia. *STAG1* encodes a subunit of cohesin, a protein complex required for correct chromosomal segregation during cell division[18]. Defects in cohesin subunits and interactors are associated with a heterogeneous class of neurodevelopmental disorders termed cohesinopathies, whose pathology is thought to be mediated by a further role of cohesin in 3D genome organisation[19]. Cohesin participates in both chromatin

**Table 2 | Published rare coding variant enrichment statistics in the novel exome-wide significant and false discovery rate <5% schizophrenia genes for additional psychiatric and developmental disorders**

| Gene | Variant Class | Schizophrenia P-value | Bipolar disorder P-value | ASD Log$_{10}$ Bayes factor | DD P-value | Epilepsy P-value |
|---|---|---|---|---|---|---|
| STAG1 | PTV | **1.9 × 10$^{-4}$** | 1.0 | 0.33 | **8.7 × 10$^{-4}$** | 0.92 |
| | Missense | **5.3 × 10$^{-5}$** | 0.058 | 0.0 | **4.6 × 10$^{-5}$** | 0.47 |
| ZNF136 | PTV | **6.2 × 10$^{-7}$** | 0.1 | 0.0 | 0.16 | 0.74 |
| | Missense | NA | 0.21 | 0.0 | 1.0 | - |
| SLC6A1 | PTV | - | - | 1.1 | **2.3 × 10$^{-8}$** | 0.16 |
| | Missense | **1.9 × 10$^{-6}$** | 0.014 | **10** | **1.0 × 10$^{-14}$** | 9.5 × 10$^{-5}$ |
| PCLO | PTV | **2.7 × 10$^{-6}$** | 0.12 | 0.44 | 0.13 | 0.34 |
| | Missense | NA | 0.058 | 0.0 | 0.31 | - |
| ZMYND11 | PTV | **1.6 × 10$^{-5}$** | 0.49 | **1.9** | **2.8 × 10$^{-9}$** | 0.72 |
| | Missense | 0.47 | 0.63 | 0.1 | **5.0 × 10$^{-9}$** | 0.20 |
| BSCL2 | PTV | **1.9 × 10$^{-5}$** | 0.72 | 0.019 | 0.14 | 0.80 |
| | Missense | NA | 0.51 | 0.0 | 0.22 | - |
| KLC1 | PTV | 0.98 | 0.50 | 0.0 | 0.18 | - |
| | Missense | **5.0 × 10$^{-5}$** | 0.62 | 0.056 | 1.0 | 0.086 |
| CGREF1 | PTV | **5.0 × 10$^{-5}$** | 1.0 | **2.9** | 0.071 | 0.27 |
| | Missense | NA | 0.24 | 0.0 | - | - |

Schizophrenia P-values are two-sided and derived from the case-control-de novo variant meta-analysis as described in the Methods. Bipolar disorder P values (two-sided Fisher Exact tests) were taken from ref. 17. ASD log$_{10}$ Bayes Factors (TADA Bayesian framework) were taken from[15]. DD/ID P values (one-sided DeNovoWEST tests) were taken from ref. 14. Epilepsy P values (two-sided Firth logistic regression tests) were taken from ref. 16. All P values in Table 2 are uncorrected for multiple comparisons. Bold P values indicate classes of mutation contributing to associations reported to be exome-wide significant or significant at a false discovery rate of 5% in the corresponding publication. Schizophrenia NA values correspond to genes without a potential MPC > 2 variant. Dashes ("-") correspond to either genes not tested in schizophrenia due to low variant numbers (see Methods), or for genes where enrichment statistics were not reported in the given study.

looping[20] and formation of topologically associating domains (TADs)[21]. Studies have shown that loss of cohesin components, including *STAG1*, is associated with disrupted patterns of chromatin contact and gene expression, including genes with functions related to neuronal development[22–24]. Previous sequencing studies of schizophrenia have provided additional evidence for dysregulated chromatin in schizophrenia, by showing cases are enriched for RCVs and de novo coding variants in sets of genes related to chromatin modification and organisation[12,25]. Moreover, schizophrenia cases carry an excess of rare structural variants disrupting TAD boundaries compared with controls[26]. By implicating *STAG1* at exome-wide significance in schizophrenia, we contribute further evidence suggesting an aetiological role for disrupted chromatin organisation in this disorder. In future studies, WGS can be used to investigate the role of TAD disrupting non-coding rare variants in schizophrenia, and also to determine whether schizophrenia is associated with disrupted TAD boundaries in specific cell types and/or developmental timepoints.

We also found that rare PTVs in *ZNF136* are enriched in schizophrenia at exome-wide significance. *ZNF136* encodes a zinc-finger protein that contains a Krüppel-associated Box (KRAB) domain, which is thought to act as a transcriptional repressor[27,28]. However, the functional roles of *ZNF136* are not well characterised. Unlike all remaining genes currently shown to be enriched for PTVs in schizophrenia with exome-wide significance, *ZNF136* displays no evidence for selective constraint against PTVs (gnomAD probability of being loss-of-function intolerant = 0)[10]. Small-scale transcriptomic studies suggest that *ZNF136* is downregulated in schizophrenia[29,30], but the mechanisms by which PTVs in *ZNF136* may increase risk for schizophrenia are unclear.

It is important for exome-wide gene discovery studies to apply stringent genome-wide thresholds for statistical significance to reduce the reporting of false positives and to ensure that funding for functional follow-up studies is prioritised towards true targets[31]. *STAG1* and *ZNF136* were provisionally implicated by the SCHEMA study at FDR < 5%[12], but our findings with a larger sample indicate that these genes reach exome-wide significance after Bonferroni correction

for multiple testing. Other genes implicated in schizophrenia by the SCHEMA study, which did not achieve exome-wide significance after Bonferroni correction but which passed the FDR < 5% threshold, have also been implicated with greater certainty in larger samples[13]. In this regard, we also identified 6 additional genes associated with schizophrenia for the first time at FDR < 5% (*SLC6A1*, *PCLO*, *ZMYND11*, *BSCL2*, *KLC1* and *CGREF1*). *SLC6A1* and *KLC1* are the first genes to be implicated in schizophrenia at FDR < 5% by missense variants (MPC > 2) alone. Sequencing data included in the published SCHEMA study supports association between damaging missense variants (MPC > 2) in these genes and schizophrenia[12], but the SCHEMA analysis down-weighted association statistics for missense MPC 2-3 variants relative to PTVs and missense MPC > 3 variants, resulting in lower power to detect associations of this class[12]. *SLC6A1* encodes a gamma-aminobutyric acid (GABA) transporter (GAT-1), which is highly expressed in GABAergic neurons and mediates uptake of GABA from the synaptic cleft of inhibitory synapses. Recently published in vitro GABA uptake assay data provides evidence that some of the published schizophrenia *SLC6A1* missense variants, as well as two of the three missense variants reported in the new cases, confer loss-of-function effects on GAT-1 protein leading to reduced GABA uptake[32]. Thus, our study supports the hypothesis of haploinsufficiency being the disease mechanism underlying risk for schizophrenia from missense variants in *SLC6A1*. The other missense variant enriched gene, *KLC1*, encodes a light chain subunit of kinesin, a tetrameric protein complex responsible for intracellular transport along the cytoskeleton. Common schizophrenia risk alleles at the *KLC1* locus are associated with reduced expression of *KLC1* RNA transcripts in the human fetal brain[33,34], and knockdown of *KLC1* has been found to impair neuronal differentiation[35]; however, the functional impact of *KLC1* missense variants observed in schizophrenia risk are unknown. A more detailed overview of the biological functions of the novel FDR < 5% genes reported in the current study is provided in Supplementary Note 2, and the spatio-temporal expression profiles of the novel exome-wide significant and FDR < 5% genes is provided in Supplementary Note 3.

Previous studies have identified four genes which show both fine-mapped common variant signals in schizophrenia GWAS and an excess of RCVs in cases at either exome-wide significance (*GRIN2A* and *SP4*) or FDR < 5% (*STAG1* and *FAM120A*)[4,12], providing strong evidence for their role in schizophrenia. Our study strengthens the evidence for *STAG1*. It also provides orthogonal support for the prioritisation of *KLC1* as a credible causal gene underlying a complex GWAS signal at this locus. The convergence of common and rare genetic liability in *STAG1* and *KLC1* makes them attractive targets for researchers aiming to develop animal and cellular models of rare high-risk variants, as the common allele signal implies that mechanistic insights gained from these models may have broad relevance across cases. When examining RCV enrichment in genes impacted by schizophrenia risk CNVs, an excess of PTVs in *NRXN1* was observed in cases compared with controls. This is a plausible finding, since intragenic deletions of *NRXN1* have consistently been shown to increase risk for schizophrenia[7,36]. No genes overlapping multi-genic schizophrenia CNV loci were enriched for RCVs after correction for multiple testing.

Genes enriched for RCVs in schizophrenia often exhibit pleiotropic effects for other psychiatric and developmental disorders, particularly DD and ASD[12,37]. Four of the eight novel exome-wide significant and FDR < 5% genes identified in the current study show enrichment for RCVs in large sequencing studies of DD, ASD and epilepsy, providing orthogonal support for their role in schizophrenia. *SLC6A1* has the broadest pleiotropic effects, wherein missense variants are enriched in ASD, DD and epilepsy. PTVs in *SLC6A1* are also associated with DD. Several of the schizophrenia genes reported in the current study also demonstrate association with syndromic neurodevelopmental disorders; for example, PTVs, missense variants, and deletions in *STAG1* cause a syndromic cohesinopathy characterised by developmental delay and mild dysmorphic features, sometimes accompanied by autistic traits and epilepsy[38–40]. Homozygous PTVs in *PCLO* are associated with pontocerebellar hypoplasia type III[41], a cause of global developmental delay and seizures. Furthermore, loss and gain-of-function mutations in *BSCL2* have been implicated in lipody-strophy and neuropathic conditions, respectively[42]. While many of the schizophrenia genes reported here show evidence of genic pleiotropy across psychiatric and developmental disorders, this does not imply functional or mechanistic overlap between these disorders, since different variants in the same gene can have distinct functional effects. Previous studies have provided evidence for pleiotropic effects from individual RCVs across schizophrenia, autism and developmental disorders[37,43], however, demonstrating allelic pleiotropy for the schizophrenia genes reported here is beyond the scope of the current study. Future studies should also determine whether particular clinical features, including neurodevelopmental phenotypes, are enriched among schizophrenia cases carrying mutations in these pleiotropic genes.

Gene-set analysis in the new sample of genes previously implicated in schizophrenia at exome-wide significance confirmed this set of genes is enriched for rare PTVs in cases compared with controls. While single-gene analysis in the new sample was underpowered, *SETD1A*, *XPO4* and *SRRM2* were enriched for RCVs in the new cases at nominal significance ($P < 0.05$). However, both our own study and a previous targeted sequencing study[13] found higher rates of rare PTVs and damaging missense variants in *CACNA1G* in controls than in cases, suggesting further work is required to determine whether *CACNA1G* is a true schizophrenia risk gene. In the new sample, we found the rate of synonymous singleton variants in constrained genes to be lower in cases compared with controls. In the context of the low coding de novo mutation rate and a disorder for which damaging de novo coding variants are a risk factor, ascertainment of probands with that disorder will be enriched for those where the random occurrence of a de novo variant is a damaging nonsynonymous mutation rather than a synonymous one. The underrepresentation of singleton synonymous variants in the new cases may reflect this ascertainment bias rather than a true protective effect of synonymous variation. However, the large SCHEMA case-control analysis did not observe any difference in the rate of rare synonymous variants in constrained genes between cases and controls, and therefore a depletion of these variants in the new cases may also be a chance finding.

A strength of our study is the large number of newly exome-sequenced cases, which we meta-analyse with published data to increase power for gene discovery. Additionally, the inclusion of a missense only variant test identified two novel genes at FDR < 5% significance. Our study also has limitations. We lack deep and longitudinal phenotype data for most of the new and published cases included in our analysis, and we are therefore unable to determine whether variants in the novel genes reported are associated with particular clinical features. This limitation can be addressed by high-quality case reports for individuals carrying mutations in the genes implicated in our study, or by genomic studies with access to linked electronic healthcare data. Moreover, we are underpowered to analyse all genetically inferred population groups in the new sample. Increasing the diversity of sequenced samples in schizophrenia will both facilitate genomic discovery and ensure more equitable progress in precision psychiatry.

In conclusion, our study implicates *STAG1* and *ZNF136* in schizophrenia with exome-wide significance and 6 additional genes at FDR < 5%. Many of these genes are enriched for RCVs in DD, ASD, and epilepsy, which supports their association with schizophrenia given the known genetic overlap between these disorders. We strengthen the evidence for an allelic series of common and rare schizophrenia risk alleles in *STAG1*, and provide evidence for the convergence of common and rare risk alleles in *KLC1*. Association of *STAG1* at exome-wide significance provides further support for an aetiological role of disrupted chromatin organisation in schizophrenia, while association of *SLC6A1* at FDR < 5% furthers the evidence implicating perturbed GABAergic neuronal signalling in the disorder.

## Methods
### Ethics statements
All research conducted as part of this study was consistent with UK regulatory and ethical guidelines. We gained National Health Service research ethics committee approval for the CLOZUK (10/WSE02/15) and Cardiff COGS (07/WSE03/110) studies. Multicentre and Local Research Ethics Committee approval was obtained for Cardiff F-Series, and all participants gave written informed consent to participate. For Cardiff Affected-Sib samples, written consent was obtained following local ethical approval guidelines. The control samples were recruited as part of independent projects, all of which have equivalent ethical permissions and data sharing procedures in place. Ethical approval for the NCMH sample was obtained from Wales Research Ethics Committee 2 (reference: 15/WA/0323). REC numbers for the Cardiff AD project are 04/9/030 and 17/SS/0139.

### New case-control exome-sequencing sample
A total of 5525 blood-derived DNA samples from schizophrenia cases were selected for exome sequencing. The majority of cases ($n = 4482$) were from the treatment-resistant schizophrenia CLOZUK cohort[44]. The remaining 1043 cases came from clinically ascertained cohorts and meet DSMIV[45] or ICD10[46] criteria for schizophrenia or schizoaffective disorder. See Supplementary Methods for further details. The new sample consisted of 7268 controls before quality control, which were derived from the following collections: Welcome Trust Case-Control Consortium (WTCCC) 2 cohort ($n = 1595$)[47]; National Centre for Mental Health (NCMH) cohort ($n = 398$)[48]; Cardiff Alzheimer's disease cohort ($n = 5275$)[49,50]. Further details about the new control sample are included in the Supplementary Methods.

### Generation and analysis of whole-exome sequencing data

All participants in the new sample were exome-sequenced on the Illumina HiSeq 4000 platform, using the Nextera DNA Exome Capture Kit, Illumina HiSeq 3000/4000 PE Cluster Kit and Illumina HiSeq 3000/4000 SBS Kit. Processing of raw sequence reads was implemented in accordance with Genome Analysis ToolKit (GATK) Best Practice guidelines[51]. The Burrow–Wheeler Aligner (v0.7.15)[52] was used to align reads to the human reference genome (GRCh37). Variants were joint-called across all samples using the GATK Haplotype Caller V3.4 and filtered using the GATK Variant Quality Score Recalibration (VQSR) tool. Mean genotype depth was 26.0x for cases and 34.2x for controls.

### Sample, variant and genotype quality control

Quality control was performed using Hail (v0.2.60) and R (v4.2.3). Sample quality control is fully described in Supplementary Methods. Briefly, samples were excluded if they had a call rate <0.75, a mean genotype depth <10, or if their sex predicted from the sequencing data did not match their recorded sex. Related individuals were also excluded to ensure that no two samples were third-degree or closer in relationship, prioritising the retention of schizophrenia samples. Principal component analysis was used to assess and control for population structure. Finally, we excluded samples that failed one or more hard sequencing quality filters (described in the 'Hard filters' section of the Supplementary Methods), and also samples predicted to overlap with the SCHEMA study based on an identity-by-descent analysis of microarray data, or the per-sample percentage of singleton coding variants observed in SCHEMA (described in the 'Exclusion of samples overlapping with SCHEMA study' section of the Supplementary Methods). Following sample quality control, 4650 cases and 5719 controls were retained.

### Association analysis

Gene-level RCV counts from the new case-control sample were meta-analysed with those from the SCHEMA study[12], which we derived using publicly available variant-level data (see 'Analysis of SCHEMA rare coding variant data' section of the Supplementary Methods). This resulted in a total of 28,898 cases and 103,041 controls. Autosomal and pseudoautosomal genes were analysed using two-sided Cochran-Mantel-Haenszel (CMH) tests with continuity correction, with separate contingency tables for the 11 SCHEMA strata (Supplementary Table 2) and the new case-control sample. Separate contingency tables for the 11 SCHEMA strata were used to control for ancestry and sequencing technology in the SCHEMA sample. Since sex-stratified variant counts are not available from SCHEMA, association statistics for non-pseudoautosomal genes on the X chromosome were generated using two-sided CMH tests with continuity correction, with separate contingency tables for the 11 SCHEMA strata and 2 additional contingency tables for males and females in the new sample.

We analysed variants with a MAC ≤ 5 in all cases and controls (SCHEMA + new sample). Since some variants in the new sample may not have been included in the SCHEMA study for quality or technical reasons, we also excluded variants in our new sample that had a MAC > 5 in 60,146 gnomAD controls. In our full case-control meta-analysis, per-gene CMH P-values for rare synonymous variants followed a null distribution (Supplementary Fig. 10).

Single gene enrichment tests were performed for four variant classes: PTVs (18,318 genes); PTVs + missense MPC > 3 (2034 genes); PTVs + missense MPC > 2 (4991 genes); and missense MPC > 2 (4991 genes). Tests involving missense variants were limited to genes containing at least one possible missense variant with an MPC score above a given threshold (i.e., 2034 genes have ≥ 1 possible missense variant with an MPC score > 3). To mitigate type I error for genes that only included variants observed in a single stratum with a small number of cases and/or controls, we only tested genes with variants observed in at least two strata, or in the largest European stratum (n gene tests

excluded across the four variant classes = 1724); we note that these excluded tests still contribute to our multiple testing criteria. We also excluded three genes (TET2, DNMT3A, and ASXL1) known to be associated with age-related clonal haematopoiesis[53,54]. In total, 30,334 single gene tests were performed in the case-control analysis.

Similar to the approach used in the SCHEMA study, we meta-analysed genes achieving a case-control CMH P < 0.01 (n genes = 340) with gene-based de novo coding variant P-values derived from 3444 published schizophrenia trios[11] using Fisher's combined probability tests. Gene-based de novo coding variant enrichment statistics were generated using Poisson rate ratio tests[11,55]. The final test statistic reported for each gene is the minimum P-value across the following tests: (1) case-control only (30,334 tests); (2) case-control and de novo variant (340 tests). This gave a total of 30,674 tests, corresponding to an exome-wide significance threshold of $P = 1.63 \times 10^{-6}$. We note that this serves as a conservative threshold, as the tests are not independent given the overlap of variants in each test.

A gene discovery sensitivity analysis that excludes individuals with Alzheimer's disease from the new control sample is presented in Supplementary Table 10.

### Gene set analysis

Gene set enrichment analyses were conducted using Firth's logistic regression tests. Here, case-control status was regressed on the number of variants carried in a given gene-set, controlling for the first ten principal components derived from common variants in the exome-sequencing data, sex, and the exome-wide burden of rare coding variants. See Supplementary Methods for further details.

### Reporting summary

Further information on research design is available in the Nature Portfolio Reporting Summary linked to this article.

## Data availability

The genetic variants contributing to the schizophrenia risk genes reported in the current study are presented in Supplementary Data 1. Aggregated variant counts at the gene level are provided in Supplementary Data 2. All datasets included in the current study are described in the Methods and Supplementary Methods. Individuals in the new case sample did not consent to public sharing of their raw genetic data, and we do not have ethical approval to deposit their exome-sequence data in controlled access repositories such as dbGaP or the EGA. Exome-sequence data from the new cases is only available through collaboration with the relevant PI, who can be contacted via the corresponding author (E.R.). Through our collaboration with SCHEMA, exome-sequence data from the new cases have been deposited into the SCHEMA Consortium. As described in the SCHEMA paper, requests for access to the controlled SCHEMA datasets are managed by data custodians of the SCHEMA consortium and the Broad Institute and are sent to sample contributing investigators for approval (see ref. 12 for further details). Access to WTCCC2 control biological samples is managed and approved by the WTCCC. The Accession number for Sequence data from the Alzheimer's disease European sequencing project, which includes a subset of the Cardiff Alzheimer's disease cohort, is: dbGaP (phs000572.v7.p4 [https://www. ncbi.nlm.nih.gov/projects/gap/cgi-bin/study.cgi?study_id= phs000572.v7.p4] (stage 1)). SCHEMA case-control variants included in the current study are available for download through the SCHEMA browser (https://schema.broadinstitute.org/). Source data for Fig. 1 are provided as a Source Data file. Source data are provided with this paper.

## Code availability

The software and code used in this study are described in the Methods. The code used to generate the gene-set and single gene results are

available at https://github.com/sophie-chick/SZ_gene_discovery and on Zenodo at https://doi.org/10.5281/zenodo.14865529[56].

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

## Acknowledgements

This work was supported by a UKRI Future Leaders Fellowship Grant to E.R. (MR/T018712/1 and MR/Y033922/1), a MRC Programme Grant no. MR/P005748/1 to M.J.O., M.C.O'D., J.T.R.W. and P.H., a MRC Programme Grant no. MR/Y004094/1 to M.J.O., M.C.O'D., J.T.R.W., P.H., and E.R. and a Fieldrose Charitable Trust PhD scholarship to S.L.C. funded by Mental Health Research UK and the Schizophrenia Research Fund. The Cardiff Alzheimer's disease cohort was supported by an MRC programme grant, funding from Health and Care Research Wales and a charitable donation from the Moondance Foundation. We acknowledge Jun Han, Joanne Morgan, Lesley Bates, Lucinda Hopkins, Rachel Raybould, Nicola Denning, Alun Meggy and Rachel Marshall at Cardiff University, for laboratory sample management and sequencing. We also acknowledge Mark Einon, at Cardiff University, for support with the use and setup of computational infrastructures and Antonio Pardiñas at Cardiff University for advice on schizophrenia common variant analysis.

## Author contributions

E.R., P.H., J.T.R.W., M.C.O.D. and M.J.O. designed the research. J.T.R.W., M.C.O.D., M.J.O., R.S. and J.W. collected the data. E.R., D.G. and R.S. processed the raw sequence data. E.R. and S.L.C. performed QC, the statistical analysis, and prepared the manuscript. D.C. and N.J.B. generated the expression profiles of implicated genes. E.R., P.H., J.T.R.W., N.J.B., M.C.O.D., M.J.O. and S.L.C. revised the manuscript. All authors read and approved the final manuscript.

## Competing interests

E.R., J.T.R.W., M.C.O. and M.J.O. reported receiving grants from Akrivia Health outside the submitted work. J.T.R.W., N.J.B., M.J.O. and M.C.O. reported receiving grants from Takeda Pharmaceutical Company Ltd outside the submitted work. Takeda and Akrivia played no part in the conception, design, implementation, or interpretation of this study. The remaining authors declare no competing interests.
