## [Transparent Peer Review file · Nature Communications]

Whole-exome sequencing analysis identifies genes for schizophrenia

Corresponding Author: Dr Elliott Rees

Version 0:

Reviewer comments:

Reviewer #1

(Remarks to the Author)

Sophie Chick and colleagues have analysed rare coding variants in exome-sequencing data from 4,650 schizophrenia cases and 5,719 control individuals and meta-analysed with results from the SCHEMA study, to identify novel trait-associated genes. Their analysis adds two genes to the list of genes with exome-wide significant evidence of association with increased risk of schizophrenia, and six genes to the pool of genes that are highly likely (FDR<5%) to be associated. They also show that at least some of these novel genes are likely associated with other brain disorders as well, and the exome-wide evidence for STAG1 association strengthens previous evidence suggesting that disrupted chromatin organisation is involved in schizophrenia aetiology.

The process of gene discovery in schizophrenia through genome- and exome-wide association studies of common and rare variants, respectively, is slow but steady. Although the yield of this study may look small in that context, it is nonetheless an important step forward. The study is well conducted and the conclusions drawn from its results seem well founded.

We have only a few questions and suggestions:

1. The results in figure 1 suggest that rare synonymous variants are protective against schizophrenia in the new sample, i.e. there is a higher burden of such variants in controls than cases. The authors argue that this is most likely explained by the higher average read depth among controls than cases, and that “any potential bias arising from this will be towards conservative estimates of schizophrenia rare variant enrichment rather than false positive discovery” (lines 122-124). While this is most likely correct, it still reduces the credibility of the study to have a batch effect between cases and controls, which is not accounted for but rather just “explained away”. It would be more convincing (and informative) if this hypothesis/explanation were tested directly, by downsampling the control BAM files to the average read depth of the case sample and verifying that the synonymous variant ratio reaches parity. Such an added sensitivity analysis would in our view strengthen the study and allow for a more direct comparison with SCHEMA results across all categories. It shouldn't be a very time-demanding effort, as downsampling of BAM files can be done with a simple command in e.g. Samtools (and from reading the methods it looks like the variant calling follows a standard, tried-and-tested GATK pipeline).
2. In Figure 1, should the y-axis not be on the log scale? Otherwise the CI95 are assymetrical.
3. In Table 1, could the authors please include OR and P for Schema-only, so we can see the before/after contrast more clearly?
4. At the start of the discussion (line 257), it would increase clarity to specify that the joint analysis is a meta-analysis (for example say “meta-analysed” instead of “jointly analysed”).
5. In the introduction (lines 63-65) the authors mention that 12 out of 13 CNVs associated with schizophrenia involve multiple genes. Does the new meta-analysis of rare disruptive SNVs offer any indication towards “key” genes in one or more of those 12 loci? In such case, it would be interesting to see it mentioned in the manuscript.

(Remarks on code availability)

Reviewer #2

(Remarks to the Author)

(Remarks on code availability)

Reviewer #3

(Remarks to the Author)

In this study, Chick et al. analyzes whole exome data from a large-scale case-control cohort for rare coding variants associated with schizophrenia. They identify 8 genes reaching exome-wide significance, including 2 novel variants in STAG1 and ZNF136, and correlate enrichment to other developmental disorders. Results of this study substantiate the discoveries of previous whole exome analyses and provide orthogonal support for the causal role of KLC1. Based on their findings, the authors propose disruption of chromatin organization and GABAergic neuronal signaling as potential etiologies of schizophrenia.

Some specific comments:

1. While some new findings resulted from this analysis, the novelty of the overall study appears minimal within the context of prior studies cited within the manuscript. The primary advantage of this work is the addition of 4,650 schizophrenia cases and 5,719 controls to an existing cohort of 24,248 cases, 97,322 controls. This allowed the authors to detect two novel gene candidates and validate previous results, but calls into question the advantage of increasing power of thoroughly-explored avenues when other methods such as a whole genome approach may yield more promising findings.

2. The structure of the study includes a discovery and replication analysis. However, the authors include their newly generated data within their replication cohort, undermining the validity of the replication analysis. Ensuring there is no overlap between replication and discovery cohort will ensure independence, reduce risk of inflated significance, and increase generalizability of results.

3. The lack of phenotypic correlations limits the specificity and potential future clinical utility of the results of this study and future studies conducted with this cohort. The authors note the heterogeneous manifestations of schizophrenia and the importance of correlating clinical features and neurodevelopmental phenotypes with genes enriched for rare protein-truncating variants, but do not propose a plan to address this gap given the lack of longitudinal phenotype data.

4. Authors note a greater rate of singleton synonymous variants in new controls when compared to cases and cite sequence depth as a likely cause. We suggest the authors perform downsampling to control for the variance between cases and controls and to ensure variability is in fact due to a difference in minor allele frequency rather than differences in sequence coverage or inconsistencies in the analysis.

5. Pleiotropic effects shown by reported p-values in Table 2 shows at best weak associations that might be better substantiated with deeper statistical correlation methods.

6. This work, in its current form, focuses primarily on gene discovery without delving into the underlying mechanisms. To advance the impact of this research, we recommend including functional studies that can provide deeper insights into the biological processes involved.

Some minor considerations:

1. As indicated in the "Analysis of SCHEMA rare coding variant data" section of the supplementary material, both whole genome and whole exome data prepared with both Nextera and non-Nextera technologies were combined in this analysis. This may introduce batch effects within the analysis and normalization should be considered or detailed if it was performed.

2. Although authors cite a lack of significant genetic correlation between Alzheimer's disease (AD) and schizophrenia, the 5,275 control samples obtained from samples with 80% AD patients do not represent true healthy controls. Furthermore, an association is drawn between genetic results of the present study to other distinct disorders such as epilepsy, bipolar disorder, and ASD. The manuscript may benefit from deeper examination of the genetic correlations between cases and controls to ensure there is no overlap and that a true control is represented in the cohort especially given the lack of clinical phenotype information.

3. Disruption of chromatin architecture is proposed as a potential driving mechanism of schizophrenia, which suggests examination of non-coding regions of the genome as a key focus in future. Authors might provide additional comments on follow-up investigations into proposed etiologies such as disruption of TAD boundaries.

(Remarks on code availability)

The provided code was available on github, but the variant table required for downstream analyses was missing.

Reviewer #4

(Remarks to the Author)

(Remarks on code availability)

Reviewer #5

(Remarks to the Author)

In this manuscript, the authors first conducted deep exome-sequencing on 4650 schizophrenia cases and 5719 healthy controls. They then combined their results with published SCHEMA study and performed a meta-analysis. They identified two novel schizophrenia risk genes, STAG1 and ZNF136, which showed significant associations at exome-wide significance level. They also identified 6 risk genes at FDR < 5%. These findings reveal the important role of rare coding variants in schizophrenia and have important significance for the research of the pathogenesis of schizophrenia. This study also suggests that there may be abundant rare coding variants associated with schizophrenia, and it is necessary to expand the sample size to identify these potential pathogenic variants in the future. This is an interesting and important study. I have some comments on this manuscript.

1. My major concern is the novelty of the results of this study. Through using a large-scale cohort, the authors identified two risk genes that showed significant associations at exome-wide significance level. However, I noticed that both STAG1 and ZNF136 have been reported in SCHEMA. Though these two genes did not reach exome-wide significance level in SCHEMA, these two genes showed associations at a false discovery rate of <5% (Fig.2 in SCHEMA). Thus, the findings of this study only provide further support for SCHEMA.

2. In this study, the authors identified two novel schizophrenia risk genes at Bonferroni corrected P value and another 6 risk genes at FDR < 5%, the manuscript just showed two genes, why not show these new discoveries in a figure?

3. For the new discovery of schizophrenia genes, are rare coding variants or protein-truncating variants associated with these novel genes present in previous schizophrenia GWAS? If so, do they have P-values at GWAS significant levels? As showed in Figure 2, KLC1 resides within a schizophrenia GWAS locus, but the results are not clear.

4. What are functions of the genes identified in this study? In addition to GWAS, are there any other evidence support their involvement in schizophrenia? Were they dysregulated in schizophrenia cases compared with controls?

5. What is the spatiotemporal expression pattern of the identified risk genes? Are they specifically expressed in brain or specific cell types?

(Remarks on code availability)

Reviewer #6

(Remarks to the Author)

This new report by Chick et al. presents analyses of a new set of exome sequencing data from 4,650 schizophrenia cases and 5,719 controls. To maximize power for risk gene discovery, the authors combine the new dataset with data from the Schizophrenia Exome Sequencing Meta-analysis (SCHEMA) consortium for a total of 28,898 cases, 103,041 controls and 3,444 proband-parent trios. Using a conservative multiple test correction strategy, they newly implicate two genes as having excess rare damaging variants in cases compared to controls: STAG1 and ZNF136. An additional six genes were newly implicated using a less conservative multiple test correction strategy (SLC6A1, PCLO, ZMYND11, BSCL2, KLC1, CGREF1). Several secondary analyses are reported that connect the findings to studies of rare variant studies of other psychiatric conditions and to studies of common variants in schizophrenia.

This work is of high technical quality that marks a rather incremental advance for schizophrenia genetics research.

Comments:

1. The work would mark a greater contribution to the field were the data generated made publicly available. From the data availability statement, it seems this data will be contributed to the SCHEMA consortium but not made more widely available to the community. If that is incorrect, then it should be clarified.

2. It was difficult to parse out the nature of the SCHEMA data used in the study (e.g., individual genotypes vs. summary statistics). At times, the text seemed to imply individual genotypes were being used (e.g., when they stated that samples included in SCHEMA were removed from analysis) while at other times the text seemed to imply that only summary information on SCHEMA was available to them (e.g., when they stated that sex-stratified variant counts were not available

from SCHEMA). In the end, I was able to figure it out by piecing together the information in the main text, methods, and supplement. For clarity purposes, the authors might want to re-work the text a bit.

3. In the supplement, the authors stated, "Variant sites were defined as low quality and excluded if they met any of the following criteria: number of observed alleles > 6" – I was unclear on what was meant here. Is this referring to the MAC threshold used (in which case, calling these variant sites "low quality" would be the wrong descriptor)? Or, is this referring to multi-allelic sites with >5 ALT alleles? Would just clarify a bit.

4. In the supplement, it states that missense annotations were made using MPC scores as well as other approaches (e.g., Polyphen). However, I believe the main analyses only use the MPC scores. I would just clarify in the methods what those other annotation methods were used for.

(Remarks on code availability)

Version 1:

Reviewer comments:

Reviewer #1

(Remarks to the Author)

The authors have adequately addressed all my previous comments and concerns, and we have only one additional comment: We found it interesting that the finding of fewer synonymous variants among cases than controls remained significant (at least in constrained genes) in the requested sensitivity analysis. First of all this highlights the importance of not taking things for granted (in this case that this observation in the main analysis would be fully accounted for by the greater read depth among controls) and perform sensitivity analyses wherever applicable to remove doubts about interpretation of results. Also, this "protective effect" is interesting since, as the authors point out, it has been observed in other studies of neurodevelopmental disorders as well.

While this should not remove the focus away from the primary results (increased knowledge about risk genes/variants) we think the manuscript would be more complete if a sentence or two in the discussion were spent on discussing what might be behind this apparently consistent finding across studies of neurodevelopmental disorders. Does this reflect some kind of deterministic "either/or" scenario of de novo events in these genes, and if so what are the implications thereof?

(Remarks on code availability)

Reviewer #2

(Remarks to the Author)

(Remarks on code availability)

Reviewer #3

(Remarks to the Author)

Some Specific Comments:

1. The authors have addressed my critique as far as was possible, but novelty remains my strongest concern for this investigation. Their findings highlight with greater statistical stringency genes previously reported in SCHEMA through augmentation of an already large cohort.

2. As the authors assure there is no overlap between discovery and replication cohorts, I suggest revisions to the supplement and manuscript to clarify. The sections that seem to suggest overlap include excerpts beginning with line 149 and 262 of the manuscript below.

Discovery of novel schizophrenia genes

150 We performed a per-gene RCV meta-analysis using data from the new case-control sample and published variant-level data from SCHEMA cases and controls (full details in Methods and Supplementary Methods), giving a total of 28,898 cases and 103,041 controls.

Replication of previously implicated genes

263 We performed a replication analysis in the new sample of genes previously associated with RCVs in schizophrenia.

They should replace "replication analysis" with "subgroup analysis" and avoid misusing the term "replication," as they are

simply analyzing newly generated data.

3. I agree the lack of phenotypic data is an unfortunate but realistically unsolvable weakness. The authors have addressed this issue as far as is possible by suggesting potential avenues for future study.

4. The authors have addressed my concern regarding down sampling.

5. Overlap in the enrichment statistic results reported between different studies is insufficient evidence for pleiotropy. Deeper statistical methods like multivariate or Bayesian approaches can provide strong evidence for truly pleiotropic, nonspurious associations. However, I agree performing these analyses is beyond the scope and aims of this study. While the addition of lines 239-243 increases the clarity of this section, I suggest further revision of the language used in this section to define the points authors want to make, and those which they do not. The conclusion seems to center on providing evidence of overlap between their findings with other related disorders, thereby strengthening their claim of the likely contributions of their candidate genes to schizophrenia through this relationship, rather than providing evidence of functional or mechanistic overlap between the genes that would indicate pleiotropy.

6. While some mechanistic insights would more strongly distinguish this study from prior investigations, I agree functional analyses are outside the current scope and the authors have addressed potential mechanisms as far as is within the scope.

Some Minor Considerations:

1. My concern was addressed by inclusion of clarification about controls implemented for batch effects.

2. My concern here is that the exclusion and inclusion criteria for controls are not clearly defined and seem to lack a biological rationale. If the foundation for the analysis is a comparative contrast between cases and controls, the controls should be selected based on the absence of confounders such as potentially related neurological conditions. The authors are correct that inclusion of control samples with confounding disorders will reduce their power to detect novel schizophrenia-associated genes, not increase type 1 error. For this reason, I do not call into question the significance of the study's results but rather raise this concern about the clarity and cleanness of sample categorization due to its potential confounding effects which, as the authors point out, serve only to reduce the power and precision of the investigation. Currently the authors cite a single study showing lack of association between AD and schizophrenia. However, other recent studies suggest associations such as Ohi et al. 2024 The supplement will benefit from a more extensive exposition of inclusion and exclusion criteria with a biological rationale. Post-hoc removal of subsections of data to determine their effects on the results is insufficient rationale for exclusion or inclusion. (Section 6 of the Supplementary Material) This clarification is especially important because the authors go on to correlate schizophrenia with other neurological disorders in Table 2. If the authors want to make a point about these correlations, they should clarify why a line is drawn between certain neurological diseases included in the control cohort (AD), but not these disorders (ASD, epilepsy, DDs, etc.).

Ohi, K., Fujikane, D., & Shioiri, T. (2024). Genetic overlap between schizophrenia spectrum disorders and Alzheimer's disease: Current evidence and future directions - An integrative review. *Neuroscience and biobehavioral reviews*, 167, 105900. <https://doi.org/10.1016/j.neubiorev.2024.105900>

3. The authors have addressed my suggestion.

(Remarks on code availability)

The input data and scripts are available on the GitHub page above, with sufficient comments for clarity.

Reviewer #4

(Remarks to the Author)

(Remarks on code availability)

The code is available along with a README file and instructions for running the scripts.

Reviewer #5

(Remarks to the Author)

The authors did not address my major concern adequately. Considering the two genes identified in this study have been reported in a previous published paper, the significance and novelty of this study is limited. The authors should acknowledge this limitation and discuss this deeply in the manuscript. I do not see any revisions to address these important concerns.

The authors have addressed my other comments.

(Remarks on code availability)

Version 2:

Reviewer comments:

Reviewer #3

(Remarks to the Author)

In this revision, the authors have addressed my concerns sufficiently. I don't have further comments.

(Remarks on code availability)

The codes have been provided along with a README file.

Reviewer #4

(Remarks to the Author)

(Remarks on code availability)

In this revision, the authors have addressed my concerns sufficiently. I do not have further comments.

REVIEWER COMMENTS

Reviewer #1 (Remarks to the Author):

Sophie Chick and colleagues have analysed rare coding variants in exome-sequencing data from 4,650 schizophrenia cases and 5,719 control individuals and meta-analysed with results from the SCHEMA study, to identify novel trait-associated genes. Their analysis adds two genes to the list of genes with exome-wide significant evidence of association with increased risk of schizophrenia, and six genes to the pool of genes that are highly likely (FDR<5%) to be associated. They also show that at least some of these novel genes are likely associated with other brain disorders as well, and the exome-wide evidence for STAG1 association strengthens previous evidence suggesting that disrupted chromatin organisation is involved in schizophrenia aetiology.

The process of gene discovery in schizophrenia through genome- and exome-wide association studies of common and rare variants, respectively, is slow but steady. Although the yield of this study may look small in that context, it is nonetheless an important step forward. The study is well conducted and the conclusions drawn from its results seem well founded.

We have only a few questions and suggestions:

1. The results in figure 1 suggest that rare synonymous variants are protective against schizophrenia in the new sample, i.e. there is a higher burden of such variants in controls than cases. The authors argue that this is most likely explained by the higher average read depth among controls than cases, and that “any potential bias arising from this will be towards conservative estimates of schizophrenia rare variant enrichment rather than false positive discovery” (lines 122-124).

While this is most likely correct, it still reduces the credibility of the study to have a batch effect between cases and controls, which is not accounted for but rather just “explained away”. It would be more convincing (and informative) if this hypothesis/explanation were tested directly, by downsampling the control BAM files to the average read depth of the case sample and verifying that the synonymous variant ratio reaches parity.

Such an added sensitivity analysis would in our view strengthen the study and allow for a more direct comparison with SCHEMA results across all categories. It shouldn't be a very time-demanding effort, as downsampling of BAM files can be done with a simple command in e.g. Samtools (and from reading the methods it looks like the variant calling follows a standard, tried-and-tested GATK pipeline).

We have now performed the suggested downsampling sensitivity analysis, which has allowed us to examine synonymous singleton variants in a dataset where the new cases are matched to the new controls for sequencing coverage. We approached this analysis by taking advantage of the existence of BAM files that were produced using wave 1 sequencing of the relevant control cohort, where sequencing was done in the same lab

as, and contemporaneously with, our case sample. After applying the same QC procedures used in our original analysis, the coverage matched dataset consists of 4,757 cases (mean genotype depth = 27.5x) and 4,881 controls (mean genotype depth = 27.7x); we note there are fewer controls here compared with the original analysis because wave 1 sequencing is only available for 82% of samples in the Cardiff Alzheimer’s disease cohort. The 107 additional cases in the coverage matched dataset are due to differences in QC thresholds that are determined by the data (e.g. principal component thresholds and hard filter thresholds (Supplementary Figures 2 and 3).

In the coverage matched dataset, we find no difference between cases and controls for synonymous singletons in non-constrained genes (Response Table 1), but the magnitude of excess of synonymous singletons in constrained genes is unchanged when compared with the original dataset (Response Table 1). Since non-constrained genes account for 84% of all genes tested, observing an equivalent rate of synonymous singletons in these genes suggests an absence of batch effects in the coverage matched dataset. As an aside, we note that the observation of a depletion in cases for singleton synonymous variation within constrained genes has been reported before, including in a recent large sequencing paper from the bipolar exome sequencing consortium (see Figure 1 of Palmer et al 2022 (Palmer et al. 2022)). Moreover, using modest sized ADHD and ASD samples, SCHEMA also found a trend for a depletion of ultra-rare synonymous variants in constrained genes in cases (see Extended Data Figure 3 in Singh et al 2022 (Singh et al. 2022)).

Variant class	Analysis	Constrained genes (n = 3,051)		Non-constrained genes (n = 15,605)	
		OR (95% CI)	P	OR (95% CI)	P
Synonymous singletons	Original dataset	0.93 (0.89 - 0.96)	2.5 x 10 ⁻⁵	0.97 (0.95 - 0.99)	0.011
	Coverage matched dataset	0.93 (0.89 - 0.97)	2.2 x 10 ⁻⁴	0.99 (0.97 - 1.01)	0.36

Response Table 1. Gene-set analysis of synonymous singleton variants in the original dataset and in the coverage matched dataset. Odds ratios (OR) and P values are derived from Firth’s logistic regression models, covarying for 10 principal components and sex.

We then tested damaging classes of rare coding variant in the coverage matched dataset, and found slightly stronger odds ratios for PTVs and missense (MPC > 3) variants in constrained genes when compared to results obtained in the original dataset (Response Table 2). These findings support our argument that the higher control sequence coverage in the original dataset leads to conservative estimates of schizophrenia rare variant enrichment, rather than false positive discovery.

Analysis	Variant class	Original dataset		Coverage matched dataset	
		OR (95% CI)	P	OR (95% CI)	P
Constrained genes (n = 3,051)	PTV	1.17 (1.05 – 1.30)	0.0034	1.19 (1.06 – 1.35)	0.0040
	Missense MPC > 3	1.63 (1.11 – 2.40)	0.012	1.83 (1.20 – 2.80)	0.0052
	Missense MPC 2-3	1.06 (0.97 – 1.17)	0.21	1.02 (0.92 – 1.14)	0.68
Non-constrained genes (n = 15,605)	PTV	0.98 (0.94 – 1.01)	0.20	0.98 (0.94 – 1.02)	0.40
	Missense MPC > 3	0.90 (0.57 – 1.38)	0.62	0.71 (0.42 – 1.17)	0.18
	Missense MPC 2-3	1.0 (0.90 – 1.12)	0.95	1.0 (0.89 – 1.14)	0.92

Response Table 2. Gene-set analysis of constrained and non-constrained genes in the original dataset and in the coverage matched dataset. Odds ratios (OR) and P values are derived from Firth’s logistic regression models. Tests of constrained genes covary for 10 principal components, sex and total number of singleton variants exome-wide. Tests of non-constrained genes covary for 10 principal components and sex.

We also note that a key difference between our original analysis and SCHEMA is that SCHEMA adjusted their gene-set analyses for the total number of singleton coding variants (of any class) in the genome, whereas we adjusted only for the total number of singleton synonymous variants when testing PTVs and missense variants. To provide a more direct comparison with SCHEMA, we now include the same adjustments they made. When using this adjustment, we see a slightly increased excess of singleton PTVs and singleton missense variants with MPC scores > 3 in constrained genes when compared with our original analysis (Response Table 3).

Analysis	Variant class	Covaried for synonymous singletons (Previous analysis)		Covaried for all singletons (New analysis)	
		OR (95% CI)	P	OR (95% CI)	P
Constrained genes (n = 3,051)	PTV	1.14 (1.03-1.27)	0.011	1.17 (1.05 – 1.30)	0.0034
	MPC3	1.59 (1.08-2.33)	0.018	1.63 (1.11 – 2.40)	0.012
	MPC 2-3	1.04 (0.94-1.14)	0.45	1.06 (0.97 – 1.17)	0.21

Response Table 3. Comparison of gene-set enrichment results when covarying for either all synonymous singleton variants or all singleton coding variants in the original dataset.

Based on the above findings, we have made the following changes to the revised manuscript. We have updated our gene-set analysis of damaging classes of RCVs in constrained genes to control for the exome-wide rate of all coding singleton variants in the new dataset, so it is a more direct comparison with SCHEMA. We have also added

the results from Response Tables 1 and 2 to Section 2 of the Supplementary Material, and the following text describing the coverage matched sensitivity analysis to the results section (lines 118-137).

“The rate of singleton synonymous variants in constrained genes was significantly higher in our new controls than in cases (Figure 1, Supplementary Table 4). In non-constrained genes, the rates of singleton PTVs and damaging missense variants (MPC > 3 and MPC 2-3) did not differ between cases and controls, but controls were still enriched for singleton synonymous variants (OR (95% confidence interval (CI)) = 0.97 (0.95 - 0.99); P = 0.012). We tested whether the excess of singleton synonymous variants in controls is explained by greater sequencing depth in our controls (mean genotype depth = 34.2x) compared with cases (mean genotype depth = 26.0x). We found that sequencing depth in the new sample was positively correlated with the rate of singleton coding variants (Supplementary Material Section 2). We also performed a gene-set sensitivity analysis using a dataset where the new controls were matched for sequencing coverage with the new cases. Here, the rate of synonymous singletons in constrained genes remained higher in controls than in cases, but the rate of synonymous singletons in non-constrained genes did not differ between cases and controls (OR (95% CI) = 0.99 (0.97 – 1.01); P = 0.36). Moreover, the excess in cases of singleton PTVs and missense variants with MPC scores > 3 in constrained genes was greater in the coverage matched analysis when compared with the original analysis (Supplementary Table 4), suggesting that the higher control sequence coverage in the original dataset leads to conservative estimates of schizophrenia rare variant enrichment, rather than false positive discovery.”

2. In Figure 1, should the y-axis not be on the log scale? Otherwise the CI95 are assymetrical.

We agree with the reviewer and have updated Figure 1 so that the y-axis is on the log scale.

3. In Table 1, could the authors please include OR and P for Schema-only, so we can see the before/after contrast more clearly?

We have now added odds ratios and P values from our re-analysis of the SCHEMA case-control data to Table 1.

4. At the start of the discussion (line 257), it would increase clarity to specify that the joint analysis is a meta-analysis (for example say “meta-analysed” instead of “jointly analysed”).

We have made this change throughout the revised manuscript.

5. In the introduction (lines 63-65) the authors mention that 12 out of 13 CNVs associated with schizophrenia involve multiple genes. Does the new meta-analysis of rare disruptive SNVs offer any indication towards “key” genes in one or more of those 12 loci? In such case, it would be interesting to see it mentioned in the

manuscript.

We are grateful for this suggestion and have examined this. We find 5 CNV genes enriched at nominal levels of significance (Response Table 4). The most significant result is for PTVs in *NRXN1*, which is associated with non-recurrent single-gene deletions in schizophrenia. While the reviewer specifically asks about genes in multi-genic schizophrenia CNVs, we think this is an important observation to add to our manuscript. The remaining 4 genes overlap distinct multi-genic schizophrenia CNV loci (Response Table 4), but due to the strength of enrichment observed for RCVs in these genes, we cannot conclude they are the causal genes underlying the CNV.

In the revised manuscript, we have changed the “Overlap of novel genes with schizophrenia common variant loci” results section to “Convergence of RCV-enriched genes in schizophrenia common variant and CNV loci”. We have also added Response Table 4 to the Supplementary Material and the following text to the results section (lines 224-235):

“For genes overlapping the critical regions of previously implicated schizophrenia CNV loci (see Supplementary Table 5 for CNV loci), five were enriched for RCVs with nominal levels of significance ($P_{\text{uncorrected}} < 0.05$) (Supplementary Table 6). Here, the most significant RCV association was observed for PTVs in *NRXN1*, which is associated with non-recurrent single-gene deletions in schizophrenia. This association survived Bonferroni correction for the number of gene tests at this CNV locus ($n = 3$; $P_{\text{corrected}} = 0.00087$). Among multigenic schizophrenia CNV loci, the most significant RCV association was observed for PTVs in *C22orf39*, which overlaps the 22q11.2 deletion locus. Since none of the genes overlapping multi-genic schizophrenia CNVs were enriched for RCVs after Bonferroni correction for the number of gene tests in the given CNV locus (Supplementary Table 6), further evidence is needed to identify those with causal effects.”

We have also added the following text to the discussion on lines 359-363.

“When examining RCV enrichment in genes impacted by schizophrenia risk CNVs, a significant excess of PTVs in *NRXN1* was observed in cases compared with controls. This is a plausible finding, since intragenic deletions of *NRXN1* have consistently been shown to increase risk for schizophrenia. No genes overlapping multi-genic schizophrenia CNV loci were enriched for RCVs after correction for multiple testing”.

CNV locus (N gene tests)	Single gene RCV enrichment statistics				
	Gene symbol	Gene locus	Variant type	P value (uncorrected)	Odds ratio
NRXN1 del (3)	NRXN1	2:50145643-51259674	PTV	0.000291*	5.9 (2.2, 15.8)
22q11.2 del (70)	C22orf39	22:19338891-19435755	PTV	0.00384	21.1 (2, 222.9)
3q29 del (38)	UBXN7	3:196074533-196159345	PTV + MPC >3	0.00492	15.8 (2.2, 115.7)
WBS dup (41)	LIMK1	7:73497263-73536855	PTV	0.0146	Inf (NA, NA)
16p11.2 dup (46)	TAOK2	16:29984962-30003582	PTV + MPC >2	0.0148	2.1 (1.2, 3.6)

Response Table 4. Rare coding variant enriched genes in schizophrenia CNV loci.

Genes are shown if they were enriched in the meta-analysis for rare coding variants with an uncorrected P value < 0.05 and overlapped a known schizophrenia CNV locus (listed in Supplementary Table 5). “N gene tests” gives the total number of tests for a given CNV locus (Σ genes \times mutation classes tested). * indicates P values that survive Bonferroni correction for the number of genes tests in the given CNV locus. RCV = rare coding variant; CI = confidence interval; CNV = copy number variant; del = deletion; dup = duplication; Inf = infinity. Gene locus coordinates are in build 37/hg19.

Reviewer #2 (Remarks to the Author):

Reviewer #3 (Remarks to the Author):

*In this study, Chick et al. analyzes whole exome data from a large-scale case-control cohort for rare coding variants associated with schizophrenia. They identify 8 genes reaching exome-wide significance, including 2 novel variants in *STAG1* and *ZNF136*, and correlate enrichment to other developmental disorders. Results of this study substantiate the discoveries of previous whole exome analyses and provide orthogonal support for the causal role of *KLC1*. Based on their findings, the authors propose disruption of chromatin organization and GABAergic neuronal signaling as potential etiologies of schizophrenia.*

Some specific comments:

1. While some new findings resulted from this analysis, the novelty of the overall study appears minimal within the context of prior studies cited within the manuscript. The primary advantage of this work is the addition of 4,650 schizophrenia cases and 5,719 controls to an existing cohort of 24,248 cases, 97,322 controls. This allowed the authors to detect two novel gene candidates and validate previous results, but calls into question the advantage of increasing power of thoroughly-explored avenues when other methods such as a whole genome

approach may yield more promising findings.

We respectfully disagree that our study “***calls into question the advantage of increasing power of thoroughly-explored avenues when other methods such as a whole genome approach may yield more promising findings***”. Exome sequencing in schizophrenia is not a thoroughly explored avenue any more than GWAS was when many commentators were making analogous suggestions when they were relatively underpowered, but GWAS in psychiatry alone has delivered 1000s of findings that have made major impacts on the field. The rarity of rare coding variants (and other forms of alleles of large effect) means even larger samples than GWAS will ultimately be required for sequencing (and that is true for WGS), but identifying additional rare coding variant enriched genes is a key priority for the field, as unlike WGS where the additional information largely comes from intergenic and other non-coding elements, exome sequencing pinpoints specific genes. Undoubtedly, WGS will provide insights and indeed we are pursuing that approach, but at present, it is much more expensive than whole-exome sequencing, and prioritising larger exome-sequencing samples over smaller WGS samples has been empirically demonstrated to yield more discoveries in genetic association analyses (Gaynor et al. 2024).

2. The structure of the study includes a discovery and replication analysis. However, the authors include their newly generated data within their replication cohort, undermining the validity of the replication analysis. Ensuring there is no overlap between replication and discovery cohort will ensure independence, reduce risk of inflated significance, and increase generalizability of results.

There is some confusion here. We assure the reviewer that we did not include newly generated data in a discovery and once again in any replication analysis. Our primary results maximised statistical power by meta-analysing the new sample with previously published data. This analysis applied a stringent threshold for statistical significance based on exome-wide correction of all genes and classes of mutation tested. What we did additionally do as supplementary material (Supplementary Tables 7 and 9 in the revised manuscript) was report for interest the results in the new sample alone for genes previously reported in studies independent of our new dataset. That presentation certainly did not use our new data in both discovery and in replication.

3. The lack of phenotypic correlations limits the specificity and potential future clinical utility of the results of this study and future studies conducted with this cohort. The authors note the heterogenous manifestations of schizophrenia and the importance of correlating clinical features and neurodevelopmental phenotypes with genes enriched for rare protein-truncating variants, but do not propose a plan to address this gap given the lack of longitudinal phenotype data.

We agree that phenotypic information is important, but the cost of collecting detailed phenotypic evaluations of samples that are large enough for gene discovery is enormous, which is why gene discovery so far has largely relied on simple case-control status. This is the case for GWAS (Trubetskoy et al. 2022), the SCHEMA study (Singh et al. 2022) and the highly influential CNV studies (Marshall et al. 2017). Ours is a research

study and not a future directions opinion piece, but we have in review articles and opinion pieces outlined the need for more phenotypes and possible routes forward (e.g. (Owen et al. 2023; Pardiñas, Owen, and Walters 2021). Nevertheless, we have edited the following text to the revised discussion (lines 396-402)

“We lack deep and longitudinal phenotype data for most of the new and published cases included in our analysis, and we are unable to determine whether variants in the novel genes reported are associated with particular clinical features. This limitation can be addressed by high-quality case reports for individuals carrying mutations in the genes implicated in our study, or by genomic studies with access to linked electronic healthcare data.”

4. Authors note a greater rate of singleton synonymous variants in new controls when compared to cases and cite sequence depth as a likely cause. We suggest the authors perform downsampling to control for the variance between cases and controls and to ensure variability is in fact due to a difference in minor allele frequency rather than differences in sequence coverage or inconsistencies in the analysis.

We have addressed this comment in response 1 by reviewer 1.

5. Pleiotropic effects shown by reported p-values in Table 2 shows at best weak associations that might be better substantiated with deeper statistical correlation methods.

We do not believe a deeper examination of pleiotropic effects would enhance our study for the following reasons. A body of literature already exists demonstrating pleiotropic effects of rare coding variants between schizophrenia and other psychiatric and developmental disorders (Rees et al. 2021; Singh et al. 2022; Palmer et al. 2022), and we did not aim to contribute new knowledge in this area. As outlined in our conclusion, we instead present evidence for pleiotropy for the limited set of novel genes as additional support their roles in schizophrenia, given the known genetic overlap between the disorders examined. However, we acknowledge that this distinction was unclear in our paper, and we have added the rationale for examining pleiotropic effects in the novel genes to the revised results (lines 239-243).

“Genes enriched for RCVs are known to overlap between schizophrenia and other psychiatric and developmental disorders. We therefore examined whether the novel genes reported here are associated with RCVs in published studies of DD (Kaplanis et al. 2020), autism spectrum disorder (ASD) (Fu et al. 2022), epilepsy (Epi25 Collaborative et al. 2023) and bipolar disorder (BD) (Palmer et al. 2022), as this would provide further support for their role in schizophrenia.”

With regards to the strength of evidence for genic pleiotropy, we believe this is strong for ASD, DD and epilepsy, but agree it is weak for bipolar disorder. To address this point, we have removed all text from our manuscript that reported weak evidence for RCVs in the novel genes being associated with bipolar disorder.

6. This work, in its current form, focuses primarily on gene discovery without delving into the underlying mechanisms. To advance the impact of this research, we recommend including functional studies that can provide deeper insights into the biological processes involved.

We agree that our findings are of high interest for the mechanistic research community, and in our manuscript, we make this point specifically for *STAG1* and *KLC1* (lines 355-359). However, functional experiments for novel gene discoveries are not within the scope of our study, as is typical for large-scale genome-wide association analyses (Singh et al. 2022; Liu et al. 2023). We also cite functional evidence from the literature suggesting haploinsufficiency is the disease mechanism most likely underlying risk for schizophrenia from missense variants in *SLC6A1* (lines 332-337).

Some minor considerations:

1. As indicated in the “Analysis of SCHEMA rare coding variant data” section of the supplementary material, both whole genome and whole exome data prepared with both Nextera and non-Nextera technologies were combined in this analysis. This may introduce batch effects within the analysis and normalization should be considered or detailed if it was performed.

We agree the different sequencing technologies included in the SCHEMA data have the potential to cause batch effects if not accounted for. Our analysis stratified SCHEMA cases and controls by 11 independent groups defined by sequencing technology and ancestry (these strata were obtained from the SCHEMA study), and rare coding variants in each stratum were meta-analysed using a Cochran-Mantel-Haenszel test with continuity correction. Using this approach, we found no inflation for per-gene P-values for synonymous variants, indicating our analysis is well-controlled for technical and ancestry effects (see Supplementary Figure 5).

We have added the following text to the revised manuscript to more clearly indicate that our meta-analysis controls for both sequencing technology and ancestry in the SCHEMA sample (lines 466-470)

“Autosomal and pseudoautosomal genes were analysed using two-sided Cochran-Mantel-Haenszel (CMH) tests with continuity correction, with separate contingency tables for the 11 SCHEMA strata (Supplementary Table 2) and the new case-control sample. Separate contingency tables for the 11 SCHEMA strata were used to control for ancestry and sequencing technology in the SCHEMA sample.”

We have also edited the text in the “Analysis of SCHEMA rare coding variant data” section of the Supplementary Material to make clear the Cochran-Mantel-Haenszel test was used to control for sequencing technology and ancestry in the SCHEMA sample.

2. Although authors cite a lack of significant genetic correlation between Alzheimer’s disease (AD) and schizophrenia, the 5,275 control samples obtained from samples with 80% AD patients do not represent true healthy controls.

Furthermore, an association is drawn between genetic results of the present study to other distinct disorders such as epilepsy, bipolar disorder, and ASD. The manuscript may benefit from deeper examination of the genetic correlations between cases and controls to ensure there is no overlap and that a true control is represented in the cohort especially given the lack of clinical phenotype information.

It is not entirely clear to us what the primary concern is, but with regards to “genetic correlations between our cases and controls”, if the reviewer is concerned that any of the new controls had one of the disorders mentioned by the reviewer (epilepsy, bipolar disorder, and ASD), the effect would be to reduce our study’s power to discover novel genes, not increasing type 1 error. If the reviewer is concerned about sample overlap and/or relatedness, then we accounted for this in our study (see “Relatedness exclusions” section of the supplementary methods). With regards to whether including samples with AD in the new control sample impacts our results, we show in Section 6 of the Supplementary Material that removing individuals with AD from our analysis does not impact the novel gene discoveries.

3. Disruption of chromatin architecture is proposed as a potential driving mechanism of schizophrenia, which suggests examination of non-coding regions of the genome as a key focus in future. Authors might provide additional comments on follow-up investigations into proposed etiologies such as disruption of TAD boundaries.

We agree and have edited the following text in the discussion to address this point (lines 302-309).

“Moreover, schizophrenia cases carry an excess of rare structural variants disrupting TAD boundaries compared with controls (Halvorsen et al. 2020). By implicating *STAG1* at exome-wide significance in schizophrenia, we contribute further evidence suggesting an aetiological role for disrupted chromatin organisation in this disorder. In future studies, WGS can be used to investigate the role of TAD disrupting non-coding rare variants in schizophrenia, and also to determine whether schizophrenia is associated with disrupted TAD boundaries in specific cell types and/or developmental timepoints.”

Reviewer #3 (Remarks on code availability):

The provided code was available on github, but the variant table required for downstream analyses was missing.

The variant table (“combined_meta_gene_table.tsv”) has now been added to the github repository.

Reviewer #4 (Remarks to the Author):

I co-reviewed this manuscript with one of the reviewers who provided the listed reports. This is part of the Nature Communications initiative to facilitate training in

peer review and to provide appropriate recognition for Early Career Researchers who co-review manuscripts.

Reviewer #5 (Remarks to the Author):

In this manuscript, the authors first conducted deep exome-sequencing on 4650 schizophrenia cases and 5719 healthy controls. They then combined their results with published SCHEMA study and performed a meta-analysis. They identified two novel schizophrenia risk genes, STAG1 and ZNF136, which showed significant associations at exome-wide significance level. They also identified 6 risk genes at FDR < 5%. These findings reveal the important role of rare coding variants in schizophrenia and have important significance for the research of the pathogenesis of schizophrenia. This study also suggests that there may be abundant rare coding variants associated with schizophrenia, and it is necessary to expand the sample size to identify these potential pathogenic variants in the future. This is an interesting and important study. I have some comments on this manuscript.

1. My major concern is the novelty of the results of this study. Through using a large-scale cohort, the authors identified two risk genes that showed significant associations at exome-wide significance level. However, I noticed that both STAG1 and ZNF136 have been reported in SCHEMA. Though these two genes did not reach exome-wide significance level in SCHEMA, these two genes showed associations at a false discovery rate of <5% (Fig.2 in SCHEMA). Thus, the findings of this study only provide further support for SCHEMA.

It is important for genome-wide genetic association studies, such as ours, to apply stringent thresholds for statistical significance to reduce the reporting of false positives and to ensure funding for functional follow-up studies is prioritised towards true targets. This is why the main focus of our paper is on *STAG1* and *ZNF136*, genes that for the first time reach exome-wide significance after Bonferroni correction for multiple testing. Since our findings are based on a meta-analysis that includes published data from the SCHEMA study, it is unsurprising that the novel genes we report show support in SCHEMA. Beyond *STAG1* and *ZNF136*, we note that we also identify 6 additional genes associated with schizophrenia for the first time at FDR < 5%, two of which (*SLC6A1* and *KLC1*) were completely missed by SCHEMA because their study did not perform a gene discovery analysis involving only missense variants.

2. In this study, the authors identified two novel schizophrenia risk genes at Bonferroni corrected P value and another 6 risk genes at FDR < 5%, the manuscript just showed two genes, why not show these new discoveries in a figure?

We present all exome-wide and FDR < 5% significant genes in a primary display item, Table 2. We believe this table is more informative than a figure, as it displays odds ratios as well as P values. However, we acknowledge that we could have named all 6 FDR < 5% genes in the abstract, which we have now done in the revised manuscript (line 37).

3. For the new discovery of schizophrenia genes, are rare coding variants or protein-truncating variants associated with these novel genes present in previous schizophrenia GWAS? If so, do they have P-values at GWAS significant levels? As showed in Figure 2, KLC1 resides within a schizophrenia GWAS locus, but the results are not clear.

The variants included in our gene discovery meta-analysis have a MAF < 0.002%, which is too rare to have been included in the schizophrenia GWAS by Trubetskoy et al 2022, which analysed variants with a MAF > 1%. *STAG1* and *KLC1* lie within genome-wide significant common variant associated loci in schizophrenia, but the Trubetskoy study did not support that the common variant associations in the vicinity of any of these genes (or indeed many of the 287 different associated loci) are likely to be explained by coding variation. To make these results more clear, we have now edited the following text in the revised manuscript (lines 202-208)

“Two novel genes, *STAG1* and *KLC1*, overlap genomic loci with genome-wide significant common variant associations in the largest schizophrenia GWAS to date (Trubetskoy et al. 2022). *STAG1* narrowly failed the conservative criteria adopted in that study for prioritising a gene as likely to be causal. *STAG1* was also reported as a likely RCV associated gene, given it was in the FDR < 5% set of the SCHEMA study (Singh et al. 2022). Given the suggestive evidence implicating this gene by two independent study designs, *STAG1* was considered by both studies to be a highly likely schizophrenia causal gene”.

4. What are functions of the genes identified in this study? In addition to GWAS, are there any other evidence support their involvement in schizophrenia? Were they dysregulated in schizophrenia cases compared with controls?

We describe the functions of the novel exome-wide significant genes in the discussion section of our manuscript, and provide a literature review of the additional FDR < 5% novel genes, including their functions, in Section 3 of the Supplementary Material. Since this section of the Supplementary Material is easily missed, we have added an additional reference to it in the results section of revised manuscript on lines 345.

As described in response to comment 5 by reviewer 3, we reported pleiotropic effects for rare coding variants in the novel genes in disorders known to share genetic liability with schizophrenia, as this provides additional support for their roles in schizophrenia. We have now described this more clearly in the revised manuscript (see response to comment 5 by reviewer 3).

5. What is the spatiotemporal expression pattern of the identified risk genes? Are they specifically expressed in brain or specific cell types?

We thank the reviewer for this suggestion. We have now added an additional section to the Supplementary Material (Section 4) that presents the expression of each implicated gene a) across human tissues, b) in human brain regions from gestation to late

adulthood and c) in individual cell populations of the fetal and adult human brain.

Reviewer #6 (Remarks to the Author):

This new report by Chick et al. presents analyses of a new set of exome sequencing data from 4,650 schizophrenia cases and 5,719 controls. To maximize power for risk gene discovery, the authors combine the new dataset with data from the Schizophrenia Exome Sequencing Meta-analysis (SCHEMA) consortium for a total of 28,898 cases, 103,041 controls and 3,444 proband-parent trios. Using a conservative multiple test correction strategy, they newly implicate two genes as having excess rare damaging variants in cases compared to controls: STAG1 and ZNF136. An additional six genes were newly implicated using a less conservative multiple test correction strategy (SLC6A1, PCLO, ZMYND11, BSCL2, KLC1, CGREF1). Several secondary analyses are reported that connect the findings to studies of rare variant studies of other psychiatric conditions and to studies of common variants in schizophrenia.

This work is of high technical quality that marks a rather incremental advance for schizophrenia genetics research.

Comments:

1. The work would mark a greater contribution to the field were the data generated made publicly available. From the data availability statement, it seems this data will be contributed to the SCHEMA consortium but not made more widely available to the community. If that is incorrect, then it should be clarified.

Individuals in the new case sample did not provide consent for their sequence data to be made publicly available, and depositing their data into the NCBI Sequence Read Archive (or a similar resource) would violate our ethical and legal requirements. The reviewer is correct in that we have already contributed our new data to the SCHEMA consortium, which our ethics permits since this is a direct collaboration between Cardiff University and the SCHEMA consortium. To allow more researchers to access the new case exome sequencing data within a framework that abides by our ethical requirements, we have edited the following text in our data availability statement

“The genetic variants contributing to the novel schizophrenia genes reported in the current study are presented in Supplementary Data 1. Aggregated variant counts at the gene level are provided in Supplementary Data 2. All datasets included in the current study are described in the Methods and Supplementary Material. Exome-sequence data from the new cases have also been deposited into the SCHEMA consortium. *Exome-sequence data from the new cases are not publicly available as individuals in this dataset did not consent to public sharing of their raw genetic data. Exome-sequence data from the new cases is available through collaboration with the relevant PI, who can be contacted via the corresponding author (E.R).* Access to WTCCC2 control biological samples is managed and approved by the WTCCC (<https://www.wtccc.org.uk/cc2/>). Accession numbers for Sequence data from the Alzheimer’s sequencing project

included in the new control sample are: dbGaP (phs000572.v7.p4 (stage 1)) and ADSP NIAGADS: <https://dss.niagads.org/datasets/ng00067-v2/> (stage 2). SCHEMA case-control variants included in the current study are available for download through the SCHEMA browser (<https://schema.broadinstitute.org/>).”

2. It was difficult to parse out the nature of the SCHEMA data used in the study (e.g., individual genotypes vs. summary statistics). At times, the text seemed to imply individual genotypes were being used (e.g., when they stated that samples included in SCHEMA were removed from analysis) while at other times the text seemed to imply that only summary information on SCHEMA was available to them (e.g., when they stated that sex-stratified variant counts were not available from SCHEMA). In the end, I was able to figure it out by piecing together the information in the main text, methods, and supplement. For clarity purposes, the authors might want to re-work the text a bit.

We thank the reviewer for highlighting this opportunity to improve the clarity of our methods. We have revised text throughout the manuscript to more clearly state that variant-level data from the SCHEMA study was used in our gene based meta-analysis. For example, the following text has been revised in the results section (lines 150-153)

“We performed a per-gene RCV meta-analysis using data from the new case-control sample and published variant-level data from SCHEMA cases and controls (full details in Methods and Supplementary Methods), giving a total of 28,898 cases and 103,041 controls.”

And the following text has been revised in the methods section (lines 462-465).

“Gene-level RCV counts from the new case-control sample were meta-analysed with those from the SCHEMA study, which we derived using publicly available variant-level data (see ‘Analysis of SCHEMA rare coding variant data’ section of the Supplemental methods).”

We have also edited the following text in the Supplementary material:

“Additionally, for each new case/control sample with a rare variant in the 12 previously implicated exome-wide significant schizophrenia genes, or in any of the novel risk genes identified in the current study, we examined the percentage of singleton coding variants carried across the exome that are also carried by a SCHEMA case, which we determined using the published variant-level data from the SCHEMA study.”

3. In the supplement, the authors stated, “Variant sites were defined as low quality and excluded if they met any of the following criteria: number of observed alleles > 6” – I was unclear on what was meant here. Is this referring to the MAC threshold used (in which case, calling these variant sites “low quality” would be the wrong descriptor)? Or, is this referring to multi-allelic sites with >5 ALT alleles? Would just clarify a bit.

Again we thank the reviewer for highlighting this point. Here, we are referring to the number of multi-allelic sites with >5 ALT alleles. We have edited the following text in the supplementary methods:

“Variant sites were defined as low quality and excluded if they met any of the following criteria: a multi-allelic site with > 6 alternative alleles....”

4. In the supplement, it states that missense annotations were made using MPC scores as well as other approaches (e.g., Polyphen). However, I believe the main analyses only use the MPC scores. I would just clarify in the methods what those other annotation methods were used for.

The reviewer is correct in that we only used the MPC score, however Polyphen is part of the MPC score, with MPC being an abbreviation for ‘missense badness, Polyphen-2 and constraint’. In the revised supplementary material, we have edited the following text:

“Missense variants were annotated using the MPC score, which stands for ‘missense badness, Polyphen-2 and constraint’ (Samocha et al. 2017). This is a single score that combines information from orthogonal deleteriousness metrics.”

Response References

- Epi25 Collaborative, Siwei Chen, Benjamin M. Neale, and Samuel F. Berkovic. 2023. ‘Shared and Distinct Ultra-Rare Genetic Risk for Diverse Epilepsies: A Whole-Exome Sequencing Study of 54,423 Individuals across Multiple Genetic Ancestries’. medRxiv. <https://doi.org/10.1101/2023.02.22.23286310>.
- Fu, Jack M., F. Kyle Satterstrom, Minshi Peng, Harrison Brand, Ryan L. Collins, Shan Dong, Brie Wamsley, et al. 2022. ‘Rare Coding Variation Provides Insight into the Genetic Architecture and Phenotypic Context of Autism’. *Nature Genetics* 54 (9): 1320–31. <https://doi.org/10.1038/s41588-022-01104-0>.
- Gaynor, Sheila M., Tyler Joseph, Xiaodong Bai, Yuxin Zou, Boris Boutkov, Evan K. Maxwell, Olivier Delaneau, et al. 2024. ‘Yield of Genetic Association Signals from Genomes, Exomes and Imputation in the UK Biobank’. *Nature Genetics* 56 (11): 2345–51. <https://doi.org/10.1038/s41588-024-01930-4>.
- Halvorsen, Matthew, Ruth Huh, Nikolay Oskolkov, Jia Wen, Sergiu Netotea, Paola Giusti-Rodriguez, Robert Karlsson, et al. 2020. ‘Increased Burden of Ultra-Rare Structural Variants Localizing to Boundaries of Topologically Associated Domains in Schizophrenia’. *Nature Communications* 11 (1): 1842. <https://doi.org/10.1038/s41467-020-15707-w>.
- Kaplanis, Joanna, Kaitlin E. Samocha, Laurens Wiel, Zhancheng Zhang, Kevin J. Arvai, Ruth Y. Eberhardt, Giuseppe Gallone, et al. 2020. ‘Evidence for 28 Genetic Disorders Discovered by Combining Healthcare and Research Data’. *Nature* 586 (7831): 757–62. <https://doi.org/10.1038/s41586-020-2832-5>.
- Liu, Dongjing, Dara Meyer, Brian Fennessy, Claudia Feng, Esther Cheng, Jessica S. Johnson, You Jeong Park, et al. 2023. ‘Schizophrenia Risk Conferred by Rare

- Protein-Truncating Variants Is Conserved across Diverse Human Populations'. *Nature Genetics* 55 (3): 369–76. <https://doi.org/10.1038/s41588-023-01305-1>.
- Marshall, Christian R., Daniel P. Howrigan, Daniele Merico, Bhooma Thiruvahindrapuram, Wenting Wu, Douglas S. Greer, Danny Antaki, et al. 2017. 'Contribution of Copy Number Variants to Schizophrenia from a Genome-Wide Study of 41,321 Subjects'. *Nature Genetics* 49 (1): 27–35. <https://doi.org/10.1038/ng.3725>.
- Owen, Michael J., Sophie E. Legge, Elliott Rees, James T. R. Walters, and Michael C. O'Donovan. 2023. 'Genomic Findings in Schizophrenia and Their Implications'. *Molecular Psychiatry*, October, 1–10. <https://doi.org/10.1038/s41380-023-02293-8>.
- Palmer, Duncan S., Daniel P. Howrigan, Sinéad B. Chapman, Rolf Adolfsson, Nick Bass, Douglas Blackwood, Marco P. M. Boks, et al. 2022. 'Exome Sequencing in Bipolar Disorder Identifies AKAP11 as a Risk Gene Shared with Schizophrenia'. *Nature Genetics* 54 (5): 541–47. <https://doi.org/10.1038/s41588-022-01034-x>.
- Pardiñas, Antonio F., Michael J. Owen, and James T. R. Walters. 2021. 'Pharmacogenomics: A Road Ahead for Precision Medicine in Psychiatry'. *Neuron* 109 (24): 3914–29. <https://doi.org/10.1016/j.neuron.2021.09.011>.
- Rees, Elliott, Hugo D. J. Creeth, Hai-Gwo Hwu, Wei J. Chen, Ming Tsuang, Stephen J. Glatt, Romain Rey, et al. 2021. 'Schizophrenia, Autism Spectrum Disorders and Developmental Disorders Share Specific Disruptive Coding Mutations'. *Nature Communications* 12 (1): 5353. <https://doi.org/10.1038/s41467-021-25532-4>.
- Samocha, Kaitlin E., Jack A. Kosmicki, Konrad J. Karczewski, Anne H. O'Donnell-Luria, Emma Pierce-Hoffman, Daniel G. MacArthur, Benjamin M. Neale, and Mark J. Daly. 2017. 'Regional Missense Constraint Improves Variant Deleteriousness Prediction'. bioRxiv. <https://doi.org/10.1101/148353>.
- Singh, Tarjinder, Timothy Poterba, David Curtis, Huda Akil, Mariam Al Eissa, Jack D. Barchas, Nicholas Bass, et al. 2022. 'Rare Coding Variants in Ten Genes Confer Substantial Risk for Schizophrenia'. *Nature* 604 (7906): 509–16. <https://doi.org/10.1038/s41586-022-04556-w>.
- Trubetsky, Vassily, Antonio F. Pardiñas, Ting Qi, Georgia Panagiotaropoulou, Swapnil Awasthi, Tim B. Bigdeli, Julien Bryois, et al. 2022. 'Mapping Genomic Loci Implicates Genes and Synaptic Biology in Schizophrenia'. *Nature* 604 (7906): 502–8. <https://doi.org/10.1038/s41586-022-04434-5>.

REVIEWER COMMENTS

Reviewer #1 (Remarks to the Author):

The authors have adequately addressed all my previous comments and concerns, and we have only one additional comment: We found it interesting that the finding of fewer synonymous variants among cases than controls remained significant (at least in constrained genes) in the requested sensitivity analysis. First of all this highlights the importance of not taking things for granted (in this case that this observation in the main analysis would be fully accounted for by the greater read depth among controls) and perform sensitivity analyses wherever applicable to remove doubts about interpretation of results. Also, this "protective effect" is interesting since, as the authors point out, it has been observed in other studies of neurodevelopmental disorders as well. While this should not remove the focus away from the primary results (increased knowledge about risk genes/variants) we think the manuscript would be more complete if a sentence or two in the discussion were spent on discussing what might be behind this apparently consistent finding across studies of neurodevelopmental disorders. Does this reflect some kind of deterministic "either/or" scenario of de novo events in these genes, and if so what are the implications thereof?

We have now added to the following text to the discussion to address this finding (lines 409-421).

“In the new sample, we found the rate of synonymous singleton variants in constrained genes to be lower in cases compared with controls. In the context of the low coding *de novo* mutation rate and a disorder for which damaging *de novo* coding variants are a risk factor, ascertainment of probands with that disorder will be enriched for those where the random occurrence of a *de novo* variant is a damaging nonsynonymous mutation rather than a synonymous one. The underrepresentation of singleton synonymous variants in the new cases may reflect this ascertainment bias rather than a true protective effect of synonymous variation. However, the large SCHEMA case-control analysis did not observe any difference in the rate of rare synonymous variants in constrained genes between cases and controls, and therefore a depletion of these variants in the new cases may also be a chance finding.”

Reviewer #2 (Remarks to the Author):

Reviewer #3 (Remarks to the Author):

Some Specific Comments:

1. The authors have addressed my critique as far as was possible, but novelty remains my strongest concern for this investigation. Their findings highlight with greater statistical stringency genes previously reported in SCHEMA through augmentation of an already large cohort.

We have addressed a similar comment in response to reviewer 5 below.

2. As the authors assure there is no overlap between discovery and replication cohorts, I suggest revisions to the supplement and manuscript to clarify. The sections that seem to suggest overlap include excerpts beginning with line 149 and 262 of the manuscript below.

Discovery of novel schizophrenia genes

150 We performed a per-gene RCV meta-analysis using data from the new case-control sample and published variant-level data from SCHEMA cases and controls (full details in Methods and Supplementary Methods), giving a total of 28,898 cases and 103,041 controls.

Replication of previously implicated genes

263 We performed a replication analysis in the new sample of genes previously associated with RCVs in schizophrenia.

They should replace "replication analysis" with "subgroup analysis" and avoid misusing the term "replication," as they are simply analyzing newly generated data.

We have now edited the main text and supplement to more clearly state which analyses involve only the new case-control sample. We have replaced all instances of "replication analysis" with "analysis of previously implicated genes in the new case-control sample". These changes can be found on lines 264-271 and 404 in the main text, and lines 268-275 in the supplement.

3. I agree the lack of phenotypic data is an unfortunate but realistically unsolvable weakness. The authors have addressed this issue as far as is possible by suggesting potential avenues for future study.

No additional changes requested.

4. The authors have addressed my concern regarding down sampling.

No additional changes requested.

5. Overlap in the enrichment statistic results reported between different studies is insufficient evidence for pleiotropy. Deeper statistical methods like multivariate or Bayesian approaches can provide strong evidence for truly pleiotropic, nonspurious associations. However, I agree performing these analyses is beyond the scope and aims of this study. While the addition of lines 239-243 increases the clarity of this section, I suggest further revision of the language used in this section to define the points authors want to make,

and those which they do not. The conclusion seems to center on providing evidence of overlap between their findings with other related disorders, thereby strengthening their claim of the likely contributions of their candidate genes to schizophrenia through this relationship, rather than providing evidence of functional or mechanistic overlap between the genes that would indicate pleiotropy.

We agree with the points raised by the reviewer. Pleiotropy can exist in several forms, and in our study, we examine whether the novel schizophrenia genes demonstrate evidence for genic pleiotropy, whereby rare coding variants in the same gene are associated with multiple psychiatric and/or developmental disorders. As described in our manuscript (lines 239-244), genes enriched for rare coding variants are known to overlap between schizophrenia and other psychiatric and developmental disorders (Rees et al. 2021; Singh et al. 2022), and therefore evidence of genic pleiotropy in the novel genes supports their association with schizophrenia. However, we agree with the reviewer that the presence of genic pleiotropy does not imply functional or mechanistic overlap between the disorders examined, since different variants in the same gene can have different effects on gene function.

We have now revised the results section of the manuscript to more clearly state that we are examining evidence of genic pleiotropy (lines 237 and 243-244). We have also added the following text to the discussion (lines 390-397) to outline that our study does not demonstrate functional or mechanistic overlap between schizophrenia and the other psychiatric and developmental disorders examined.

“While many of the novel schizophrenia genes reported here show evidence of genic pleiotropy across psychiatric and developmental disorders, this does not imply functional or mechanistic overlap between these disorders, since different variants in the same gene can have distinct functional effects. Previous studies have provided evidence for pleiotropic effects from individual rare coding variants across schizophrenia, autism and developmental disorders (Rees et al. 2021; Singh et al. 2016), however, demonstrating allelic pleiotropy for the novel schizophrenia genes reported here is beyond the scope of the current study”.

6. While some mechanistic insights would more strongly distinguish this study from prior investigations, I agree functional analyses are outside the current scope and the authors have addressed potential mechanisms as far as is within the scope.

No additional changes requested.

Some Minor Considerations:

1. My concern was addressed by inclusion of clarification about controls implemented for batch effects.

No additional changes requested.

2. My concern here is that the exclusion and inclusion criteria for controls are

not clearly defined and seem to lack a biological rationale. If the foundation for the analysis is a comparative contrast between cases and controls, the controls should be selected based on the absence of confounders such as potentially related neurological conditions. The authors are correct that inclusion of control samples with confounding disorders will reduce their power to detect novel schizophrenia-associated genes, not increase type 1 error. For this reason, I do not call into question the significance of the study's results but rather raise this concern about the clarity and cleanness of sample categorization due to its potential confounding effects which, as the authors point out, serve only to reduce the power and precision of the investigation. Currently the authors cite a single study showing lack of association between AD and schizophrenia. However, other recent studies suggest associations such as Ohi et al. 2024 The supplement will benefit from a more extensive exposition of inclusion and exclusion criteria with a biological rationale. Post-hoc removal of subsections of data to determine their effects on the results is insufficient rationale for exclusion or inclusion. (Section 6 of the Supplementary Material)

This clarification is especially important because the authors go on to correlate schizophrenia with other neurological disorders in Table 2. If the authors want to make a point about these correlations, they should clarify why a line is drawn between certain neurological diseases included in the control cohort (AD), but not these disorders (ASD, epilepsy, DDs, etc.).

Ohi, K., Fujikane, D., & Shioiri, T. (2024). Genetic overlap between schizophrenia spectrum disorders and Alzheimer's disease: Current evidence and future directions - An integrative review. Neuroscience and biobehavioral reviews, 167, 105900. <https://doi.org/10.1016/j.neubiorev.2024.105900>

The inclusion criteria for the new control sample is based on power rather than biology, and is justified by evidence that, in genetic studies of relatively uncommon disorders, the increased power afforded by larger samples of controls, even completely unscreened controls, is expected to be more than offset by the accidental inclusion of cases in that larger control sample (Moskvina et al. 2005).

Regarding the overlap between schizophrenia and AD, the Ohi study reported evidence for very weak genetic overlap ($r^2 = 0.03-0.1$) but also acknowledged the evidence from other types of analyses (polygenic risk scoring) as inconsistent. Moreover, the PRS studies referred to examined the impact of SZ-PRS not on liability to AD but liability to psychosis in people with AD (Ohi, Fujikane, and Shioiri 2024; Creese et al. 2019), which is not informative for the present issues. We also note that to the best of our knowledge, no studies have been published demonstrating a rare variant overlap between AD and schizophrenia, which is not the case for the others disorders the reviewer highlights (ASD, epilepsy and DD) (Rees et al. 2021; Singh et al. 2022).

To address this comment, we have now added the following text to the supplementary material (lines 58-66).

“We included individuals with AD in our new control sample given the evidence that in genetic studies of relatively uncommon disorders, the increased power afforded by

larger samples of controls, even completely unscreened controls, is expected to be more than offset by the accidental inclusion of cases (Moskvina et al. 2005). We also note that while some studies have reported a very weak positive genetic association between schizophrenia and AD (r^2 0.03-0.1) (Ohi, Fujikane, and Shioiri 2024), the evidence is inconsistent (The Brainstorm Consortium et al. 2018; Wingo et al. 2022). Moreover, to the best of our knowledge, no studies have been published demonstrating a rare variant overlap between AD and schizophrenia.”

3. The authors have addressed my suggestion.

No additional changes requested.

Reviewer #3 (Remarks on code availability):

The input data and scripts are available on the GitHub page above, with sufficient comments for clarity.

No additional changes requested.

Reviewer #4 (Remarks to the Author):

Reviewer #4 (Remarks on code availability):

The code is available along with a README file and instructions for running the scripts.

No additional changes requested.

Reviewer #5 (Remarks to the Author):

The authors did not address my major concern adequately. Considering the two genes identified in this study have been reported in a previous published paper, the significance and novelty of this study is limited. The authors should acknowledge this limitation and discuss this deeply in the manuscript. I do not see any revisions to address these important concerns.

To address this concern, we have added the following text to the discussion (lines 324-335).

“It is important for exome-wide gene discovery studies to apply stringent genome-wide thresholds for statistical significance to reduce the reporting of false positives and to ensure that funding for functional follow-up studies is prioritised towards true targets (Wang et al. 2021). *STAG1* and *ZNF136* were provisionally implicated by the SCHEMA study at $FDR < 5\%$, but our findings in a larger sample indicate that these genes reach exome-wide significance after Bonferroni correction for multiple

testing. Other genes implicated in schizophrenia by the SCHEMA study, which did not achieve exome-wide significance after Bonferroni correction but which passed the FDR < 5% threshold, have also been implicated with greater certainty in larger samples (Liu et al. 2022). In this regard, we also identified 6 additional genes associated with schizophrenia for the first time at FDR < 5% (*SLC6A1*, *PCLO*, *ZMYND11*, *BSCL2*, *KLC1* and *CGREF1*).”

The authors have addressed my other comments.

No additional changes requested.

Response references

- Creese, Byron, Evangelos Vassos, Sverre Bergh, Lavinia Athanasiu, Iskandar Johar, Arvid Rongve, Ingrid Tøndel Medbøen, et al. 2019. ‘Examining the Association between Genetic Liability for Schizophrenia and Psychotic Symptoms in Alzheimer’s Disease’. *Translational Psychiatry* 9 (1): 273. <https://doi.org/10.1038/s41398-019-0592-5>.
- Liu, Dongjing, Dara Meyer, Brian Fennessy, Claudia Feng, Esther Cheng, Jessica S. Johnson, You Jeong Park, et al. 2022. ‘Rare Schizophrenia Risk Variant Burden Is Conserved in Diverse Human Populations’. medRxiv. <https://doi.org/10.1101/2022.01.03.22268662>.
- Moskvina, V., P. Holmans, K. M. Schmidt, and N. Craddock. 2005. ‘Design of Case-Controls Studies with Unscreened Controls’. *Annals of Human Genetics* 69 (5): 566–76. <https://doi.org/10.1111/j.1529-8817.2005.00175.x>.
- Ohi, Kazutaka, Daisuke Fujikane, and Toshiki Shioiri. 2024. ‘Genetic Overlap between Schizophrenia Spectrum Disorders and Alzheimer’s Disease: Current Evidence and Future Directions - An Integrative Review’. *Neuroscience and Biobehavioral Reviews* 167 (December):105900. <https://doi.org/10.1016/j.neubiorev.2024.105900>.
- Rees, Elliott, Hugo D. J. Creeth, Hai-Gwo Hwu, Wei J. Chen, Ming Tsuang, Stephen J. Glatt, Romain Rey, et al. 2021. ‘Schizophrenia, Autism Spectrum Disorders and Developmental Disorders Share Specific Disruptive Coding Mutations’. *Nature Communications* 12 (1): 5353. <https://doi.org/10.1038/s41467-021-25532-4>.
- Singh, Tarjinder, Mitja I. Kurki, David Curtis, Shaun M. Purcell, Lucy Crooks, Jeremy McRae, Jaana Suvisaari, et al. 2016. ‘Rare Loss-of-Function Variants in SETD1A Are Associated with Schizophrenia and Developmental Disorders’. *Nat Neurosci* 19 (4): 571–77. <https://doi.org/10.1038/nn.4267>.
- Singh, Tarjinder, Timothy Poterba, David Curtis, Huda Akil, Mariam Al Eissa, Jack D Barchas, Nicholas Bass, et al. 2022. ‘Rare Coding Variants in Ten Genes Confer Substantial Risk for Schizophrenia’. *Nature* 604 (7906): 509–16.
- The Brainstorm Consortium, Verner Anttila, Brendan Bulik-Sullivan, Hilary K. Finucane, Raymond K. Walters, Jose Bras, Laramie Duncan, et al. 2018. ‘Analysis of Shared Heritability in Common Disorders of the Brain’. *Science* 360 (6395): eaap8757. <https://doi.org/10.1126/science.aap8757>.
- Wang, Shan, Anna Bleeck, Nael Nadif Kasri, Tjitske Kleefstra, Jon-Ruben van Rhijn, and Dirk Schubert. 2021. ‘SETD1A Mediated H3K4 Methylation and Its Role in Neurodevelopmental and Neuropsychiatric Disorders’. *Frontiers in*

Molecular Neuroscience 14 (November):772000.

<https://doi.org/10.3389/fnmol.2021.772000>.

Wingo, Thomas S., Yue Liu, Ekaterina S. Gerasimov, Selina M. Vattathil, Meghan E. Wynne, Jiaqi Liu, Adriana Lori, et al. 2022. 'Shared Mechanisms across the Major Psychiatric and Neurodegenerative Diseases'. *Nature Communications* 13 (1): 4314. <https://doi.org/10.1038/s41467-022-31873-5>.